# TCT: Convexifying Federated Learning using Bootstrapped Neural Tangent Kernels

**Yaodong Yu**
UC Berkeley
yyu@eecs.berkeley.edu

**Alexander Wei**
UC Berkeley
awei@berkeley.edu

**Sai Praneeth Karimireddy**
UC Berkeley
sp.karimireddy@berkeley.edu

**Yi Ma**
UC Berkeley
yima@eecs.berkeley.edu

**Michael I. Jordan**
UC Berkeley
jordan@cs.berkeley.edu

## Abstract

State-of-the-art federated learning methods can perform far worse than their centralized counterparts when clients have dissimilar data distributions. For neural networks, even when centralized SGD easily finds a solution that is simultaneously performant for all clients, current federated optimization methods fail to converge to a comparable solution. We show that this performance disparity can largely be attributed to optimization challenges presented by *nonconvexity*. Specifically, we find that the early layers of the network do learn useful features, but the final layers fail to make use of them. That is, federated optimization applied to this non-convex problem distorts the learning of the final layers. Leveraging this observation, we propose a ***T**rain-**C**onvexify-**T**rain* (TCT) procedure to sidestep this issue: first, learn features using off-the-shelf methods (e.g., FedAvg); then, optimize a *convexified* problem obtained from the network's empirical neural tangent kernel approximation. Our technique yields accuracy improvements of up to $+36\%$ on FMNIST and $+37\%$ on CIFAR10 when clients have dissimilar data.

## 1 Introduction

Federated learning is a newly emerging paradigm for machine learning where multiple data holders (clients) collaborate to train a model on their combined dataset. Clients only share partially trained models and other statistics computed from their dataset, keeping their raw data local and private [53, 37]. By obviating the need for a third party to collect and store clients' data, federated learning has several advantages over the classical, centralized paradigm [14, 31, 23]: it ensures clients' consent is tied to the specific task at hand by requiring active participation of the clients in training, confers some basic level of privacy, and has the potential to make machine learning more participatory in general [43, 36]. Further, widespread legislation of data portability and privacy requirements (such as GDPR and CCPA) might even make federated learning a necessity [59].

Collaboration among clients is most attractive when clients have very different subsets of the combined dataset (*data heterogeneity*). For example, different autonomous driving companies may only be able to collect data in weather conditions specific to their location, whereas their vehicles would need to function under all conditions. In such a scenario, it would be mutually beneficial for companies in geographically diverse locations to collaborate and share data with each other. Further, in such settings, clients are physically separated and connected by ad-hoc networks with large latencies and limited bandwidth. This is especially true when clients are edge devices such as mobile phones, IoT sensors, etc. Thus, *communication efficiency* is crucial for practical federated learning. However, it is precisely under such circumstances (large data heterogeneity and low communication) that current

36th Conference on Neural Information Processing Systems (NeurIPS 2022).

algorithms fail dramatically [27, 48, 39, 61, 71, 1, 46, 3, 72, etc.]. This motivates our central question: *Why do current federated methods fail in the face of data heterogeneity—and how can we fix them?*

**Our solution.** We make two main observations: (i) We show that, even with data heterogeneity, linear models can be trained in a federated manner through gradient correction techniques such as SCAFFOLD [39]. While this observation is promising, it alone remains limited, as linear models are not rich enough to solve practical problems of interest (e.g., those that require feature learning). (ii) We shed light on why current federated algorithms struggle to train deep, nonconvex models. We observe that the failure of existing methods for neural networks is not uniform across the layers. The early layers of the network do in fact learn useful features, but the final layers fail to make use of them. Specifically, federated optimization applied to this nonconvex problem results in distorted final layers.

These observations suggest a *train-convexify-train* federated algorithm, which we call *TCT*: first, use any off-the-shelf federated algorithm [such as FedAvg, 53] to train a deep model to extract useful features; then, compute a convex approximation of the deep model using its empirical Neural Tangent Kernel (eNTK) [34, 44, 20, 51, 75], and use gradient correction methods such as SCAFFOLD to train the final model. Effectively, the second-stage features freeze the features learned in the first stage and fit a linear model over them. We show that this simple strategy is highly performant on a variety of tasks and models—we obtain accuracy gains up to 36% points on FMNIST with a CNN, 37% points on CIFAR10 with ResNet18-GN, and 16% points on CIFAR100 with ResNet18-GN. Further, its convergence remains unaffected even by extreme data heterogeneity. Finally, we show that given a pre-trained model, our method completely closes the gap between centralized and federated methods.

## 2 Related Work

**Federated learning.** There are two main motivating scenarios for federated learning (FL). The first is where internet service companies (e.g., Google, Facebook, Apple, etc.) want to train machine learning models over their users' data, but do not want to transmit raw personalized data away from user devices [60, 8]. This is the setting of *cross-device* federated learning and is characterized by an extremely large number of unreliable clients, each of whom has very little data and the collections of data are assumed to be homogeneous [37, 10, 38, 8]. The second motivating scenario is when valuable data is split across different organizations, each of whom is either protected by privacy regulation or is simply unwilling to share their raw data. Such "data islands" are common among hospital networks, financial institutions, autonomous-vehicle companies, etc. This is known as *cross-silo* federated learning and is characterized by a few highly reliable clients, who potentially have extremely diverse data. In this work, we focus on the latter scenario.

**Metrics in FL.** FL research considers numerous metrics, such as fairness across users [55, 47, 62], formal security and privacy guarantees [9, 60, 21, 56], robustness to corrupted agents and corrupted training data [7, 64, 19, 40, 26], preventing backdoors at test time [6, 66, 69, 52], etc. While these concerns are important, the main goal of FL (and our work) is to achieve high accuracy with minimal communication [53]. Clients are typically geographically separated yet need to communicate large deep learning models over unoptimized ad-hoc networks [37]. Finally, we focus on the setting where all users are interested in training the same model over the combined dataset. This is in contrast to model-agnostic protocols [49, 58, 3] or personalized federated learning [16, 18, 78, 13, 42, 12]. Finally, we focus on minimizing the number of rounds required. Our approach can be combined with communication compression, which reduces bits sent per round [67, 4, 24, 65].

**Federated optimization.** Algorithms for FL proceed in rounds. In each round, the server sends a model to the clients, who partially train this model using their local compute and data. The clients send these partially trained models back to the server who then aggregates them, finishing a round. FedAvg [53], which is the de facto standard FL algorithm, uses SGD to perform local updates on the clients and aggregates the client models by simply averaging their parameters. Unfortunately, however, FedAvg has been observed to perform poorly when faced with data heterogeneity across the clients [27, 48, 39, 61, 71, 1, 46, 3, 72, 17, etc.]. Theoretical investigations of this phenomenon [39, 76] showed that this was a result of *gradient heterogeneity* across the clients. Consider FedAvg initialized with the globally optimal model. If this model is not also optimal for each of the clients as well, the local updates will push it away from the global optimum. Thus, convergence would require a careful tuning of hyper-parameters. To overcome this issue, SCAFFOLD [39] and FedDyn [1] propose to use control variates to correct for the biases of the individual clients akin to variance

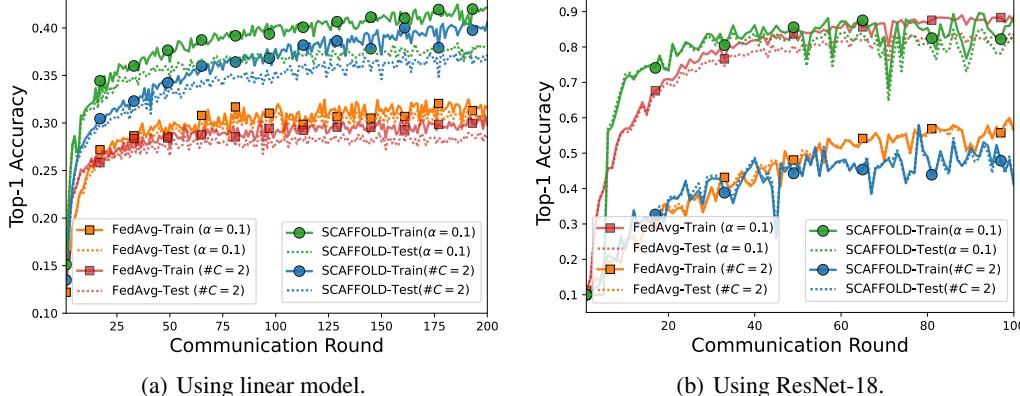

(a) Using linear model.        (b) Using ResNet-18.

Figure 1: Performance of FedAvg and SCAFFOLD on CIFAR10 when data are split among ten clients in two ways (#C=2 and $\alpha$=0.1). The #C=2 split is more non-i.i.d. than the $\alpha$=0.1 split. For convex problems (left), gradient correction methods such as SCAFFOLD are relatively unaffected by data heterogeneity, and consistently outperform FedAvg. However, for nonconvex problems (right), FedAvg and SCAFFOLD perform very similarly and both are strongly negatively affected by data heterogeneity.

reduction [35, 15]. This *gradient correction* is applied in every local update by the client and provably nullifies the effect of gradient heterogeneity [39, 54, 12]. However, as we show here, such methods are insufficient to overcome high data heterogeneity especially for deep learning. Other, more heuristic approaches to combat gradient heterogeneity include using a regularizer [48] and sophisticated server aggregation strategies such as momentum [28, 70, 50] or adaptivity [61, 38, 11].

A second line of work pins the blame on performance loss due to averaging nonconvex models. To overcome this, Singh and Jaggi [63], Yu et al. [81] propose to learn a mapping between the weights of the client models before averaging, Afonin and Karimireddy [3] advocates a functional perspective and replaces the averaging step with knowledge distillation, and Wang et al. [74], Li et al. [46], Tan et al. [68] attempt to align the internal representations of the client models. However, averaging is unlikely to be the only culprit since FedAvg does succeed under low heterogeneity, and averaging nonconvex models can lead to improved performance [33, 77].

**Neural Tangent Kernels (NTK) and neural network linearization.** NTK was first proposed to analyze the limiting behavior of infinitely wide networks [34, 44]. While NTK with MSE may be a bad approximation of real-world finite networks in general [22], it approximates the fine-tuning of a pre-trained network well [57], especially with some minor modifications [2]. That is, NTK cannot capture feature learning but does capture how a model utilizes learnt features better than last/mid layer activations.

## 3 The Effect of Nonconvexity

In this section, we investigate the poor performance of FedAvg [53] and SCAFFOLD [39] empirically in the setting of deep neural networks, focusing on image classification with a ResNet-18. To construct our federated learning setup, we split the CIFAR-10 dataset in a highly heterogeneous manner among ten clients. We either assign each client two classes (denoted by #C=2) or distribute samples according to a Dirichlet distribution with $\alpha = 0.1$ (denoted by $\alpha$=0.1). For more details, see Section 5.1.

**Insufficiency of gradient correction methods.** Current theoretical work [e.g., 39, 61, 1, 73] attributes the slowdown from data heterogeneity to the individual clients having varying local optima. If no single model is simultaneously optimal for all clients, then the updates of different clients can compete with and distort each other, leading to slow convergence. This tension is captured by the variance of the updates across the clients [client gradient heterogeneity, see 72]. Gradient correction methods such as SCAFFOLD [39] and FedDyn [1] explicitly correct for this and are provably unaffected by gradient heterogeneity for both convex and nonconvex losses.

Table 1: Feature learning by FedAvg. We report the test accuracy of a ResNet-18 after (centralized) retraining of the last $\ell$ layers on CIFAR10. The earlier $(7 - \ell)$ layers are frozen to either random initialization or to the weights of a FedAvg-trained model. The difference measures utility of the $(7 - \ell)$ layers learnt by FedAvg. The baseline FedAvg model without additional training gets 56.9% accuracy. We see that all layers of the FedAvg model contain useful information.

| Layers retrained | Accuracy (%) Random init | Accuracy (%) FedAvg init | Improvement (%) (FedAvg - Random) |
|---|---|---|---|
| 1/7 last layer | 35.37 | 77.93 | 42.56 |
| 2/7 last layers | 67.33 | 87.04 | 19.71 |
| 3/7 last layers | 80.18 | 89.28 | 9.10 |
| 4/7 last layers | 88.03 | 90.57 | 2.54 |
| 5/7 last layers | 91.34 | 91.61 | 0.27 |
| 6/7 last layers | 91.78 | 91.91 | 0.13 |

These theoretical predictions are aligned with the results of Figure 1(a), where the loss landscape is convex: SCAFFOLD is relatively unaffected by the level of heterogeneity and consistently outperforms FedAvg. In particular, performance is largely dictated by the algorithm and not the data distributions. This shows that client gradient heterogeneity captures the difficulty of the problem well. On the other hand, when training a ResNet-18 model with nonconvex loss landscape, Figure 1(b) shows that both FedAvg and SCAFFOLD suffer from data heterogeneity. This is despite the theory of gradient correction applying to both convex and nonconvex losses. Further, the train and test accuracies in Figure 1(b) match quite closely, suggesting that the failure lies in optimization (not fitting the training data) rather than generalization. Thus, while the current theory makes no qualitative distinctions between convex and nonconvex convergence, the practical behavior of algorithms in these settings is very different. Such differences between theoretical predictions and practical reality suggests that black-box notions such as gradient heterogeneity are insufficient for capturing the difficulty of training deep models.

**Ease of feature learning.**  We now dive into how a ResNet-18 trained with FedAvg (56.9% accuracy) differs from the centralized baseline (91.9% accuracy). We first apply linear probing to the FedAvg model (i.e., retraining with all but the output layer frozen). Note that this is equivalent to (convex) logistic regression over the last-layer activations. This simple procedure produces a striking jump from 56.9% to 77.9% accuracy. Thus, of the 35% gap in accuracy between the FedAvg and centralized models, 21% may be attributed to a failure to optimize the linear output layer. We next extend this experiment towards probing the information content of other layers.

Given a FedAvg-trained model, we can use centralized training to retrain only the last $\ell$ layers while keeping the rest of the $(7 - \ell)$ layers (or ResNet blocks) frozen. We can also perform this procedure starting from a randomly initialized model. The performance difference between these two models can be attributed to the information content of the frozen $(7 - \ell)$ layers of the FedAvg model. Table 1 summarizes the results of this experiment. The large difference in accuracy (up to 42.6%) indicates the initial layers of the FedAvg model have learned useful features. There continues to be a gap between the FedAvg features and random features in the earlier layers as well,[1] meaning that all layers of the FedAvg model learn useful features. We conjecture this is because from the perspective of earlier layers which perform simple edge detection, the tasks are independent of labels and the clients are i.i.d. However, the higher layers are more specialized and the effect of the heterogeneity is stronger.

## 4    Method

Based on the observations in Section 3, we propose *train-convexify-train* (TCT) as a method for overcoming data heterogeneity when training deep models in a federated setting. Our high-level

---

[1]The significant decrease in the gap as we go down the layers may be because of the skip connections in the lower ResNet blocks which allow the random frozen layers to be sidestepped. This underestimates the true utility and information content in the earlier FedAvg layers.

intuition is that we want to leverage both the features learned from applying FedAvg to neural networks and the effectiveness of *convex* federated optimization. More specifically, we perform several rounds of "*bootstrap*" FedAvg to learn features before solving a convexified version of the original optimization problem.

## 4.1 Computing the Empirical Neural Tangent Kernel

To sidestep the challenges presented by nonconvexity, we describe how we approximate a neural network by its "linearization." Given a neural network $f(\,\cdot\,;\theta_0)$ with weights $\theta_0 \in \mathbb{R}^P$ mapping inputs $x \in \mathbb{R}^D$ to $\mathbb{R}^C$, we replace it by its *empirical neural tangent kernel (eNTK)* approximation at $\theta_0$ given by

$$f(x;\theta) \approx f(x;\theta_0) + (\theta - \theta_0)^\top \frac{\partial}{\partial \theta} f(x;\theta_0),$$

at each $x \in \mathbb{R}^D$. Under this approximation, $f(x;\theta)$ is a linear function of the "feature vector" $(f(x;\theta_0), \frac{\partial}{\partial \theta} f(x;\theta_0))$ and the original nonconvex optimization problem becomes (convex) linear regression with respect to these features.[2] Leveraging NTK for solving federated optimization problems has also been studied in previous work [29, 82].

To reduce the computational burden of working with the eNTK approximation, we make two further approximations: First, we randomly reinitialize the last layer of $\theta_0$ and only consider $\frac{\partial}{\partial \theta} f(x;\theta_0)$ with respect to a single output logit. Over the randomness of this reinitialization, $\mathbb{E}[f(x;\theta_0)] = 0$. Moreover, given the random reinitialization, all the output logits of $f(x;\theta_0)$ are symmetric. These observations mean each data point $x$ can be represented by a $P$-dimensional feature vector $\frac{\partial}{\partial \theta} f_1(x;\theta_0)$, where $f_1(\,\cdot\,;\theta_0)$ refers to the first output logit. Then, we apply a dimensionality reduction by subsampling $p$ random coordinates from this $P$-dimensional featurization.[3] In our setting, this sub-sampling has the added benefit of reducing the number of bits communicated per round.

In summary, we transform our original (nonconvex) optimization problem over a neural network initialized at $\theta_0$ into a convex optimization problem in three steps: (i) reinitialize the last layer of $\theta_0$; (ii) for each data point $x$, compute the gradient $\phi_{\mathrm{eNTK}}(x;\theta_0) := \frac{\partial}{\partial \theta} f_1(x;\theta_0)$; (iii) subsample the coordinates of $\phi_{\mathrm{eNTK}}(x;\theta_0)$ for each $x$ to obtain a reduced-dimensionality eNTK representation. Let $\mathcal{S}\colon \mathbb{R}^P \to \mathbb{R}^p$ denote this subsampling operation. Finally, we solve the resulting linear regression problem over these eNTK representations.[4]

## 4.2 Convexifying Federated Learning via eNTK Representations

The eNTK approximation lets us convexify the neural net optimization problem: following Section 4.1, we may extract (from a model trained with FedAvg) eNTK representations of inputs from each client. It remains to fit an overparameterized linear model using these eNTK features in a federated manner. For ease of presentation, we denote the subsampled eNTK representation of input $x$ by $z \in \mathbb{R}^p$, where $p$ is the eNTK feature dimension after subsampling. We use $z_i^k$ to represent the eNTK feature of the $i$-th sample from the $k$-th client. Then, for $K$ the number of clients, $Y_i^k$ the one-hot encoded labels, $n_k$ the number of data points of the $k$-th client, $n := \sum_{k\in[K]} n_k$ the number of data points across all clients, and $p_k := n_k/n$, we can approximate the nonconvex neural net optimization problem by the convex linear regression problem

$$\min_W L(W) := \sum_{k=1}^K p_k \cdot L_k(W), \qquad \text{where} \quad L_k(W) := \frac{1}{n_k} \sum_{i=1}^{n_k} \|W^\top z_i^k - Y_i^k\|_2^2. \qquad (1)$$

To obtain the eNTK representation $z$ of an input $x$, we take $\theta_0$ in Section 4.1 to be the weights of a model trained with FedAvg. As we will show in Section 5, the convex reformulation in Eq. (1) significantly reduces the number of communication rounds needed to find an optimal solution.

---

[2]For classification problems, we one-hot encoded labels and fit a linear model using squared loss.

[3]That such representations empirically have low effective dimension due to fast eigenvalue decay [see, e.g., 75] means that such a random projection approximately preserves the geometry of the data points [5, 83]. For all of our experiments, we set $p = 100,000$.

[4]Given a fitted linear model with weights $W \in \mathbb{R}^{p \times C}$, the prediction at $x$ is $\arg\max_j [W^\top \mathcal{S}(\phi_{\mathrm{eNTK}}(x))]_j$.

### 4.3 Train-Convexify-Train (TCT)

We now present our algorithm train-convexify-train (TCT), with convexification done via the neural tangent kernel, for federated optimization.

---

**TCT — train-convexify-train with eNTK representations**

- **Stage 1:** *Extract eNTK features from a FedAvg-trained model.* FedAvg is first used to train the model for $T_1$ communication rounds. Let $\theta_{T_1}$ denote the model weights after these $T_1$ rounds. Then, each client locally computes subsampled eNTK features, i.e., $z_i^k = \mathcal{S}(\phi_{\mathrm{eNTK}}(x_i^k; \theta_{T_1}))$ for $k \in [K]$ and $i \in [n_k]$.
- **Stage 2:** *Decentralized linear regression with gradient correction.* Given samples $\{(z_i^k, Y_i^k)\}_{i=1}^{n_k}$ on each client $k$, first normalize the eNTK inputs of all clients with a single communication round.[a] Then, solve the linear regression problem defined in Eq. (1) by SCAFFOLD with local learning rate $\eta$ and local steps $M$.[b]

---
[a]For every feature in the eNTK representation, subtract the mean and scale to unit variance.
[b]The detailed description of SCAFFOLD for solving linear regression problems can be found in Algorithm 1, Appendix A. It has the same communication and computation cost as FedAvg.

---

To motivate TCT, recall that in Section 3 we found that FedAvg learns "useful" features despite its poor performance, especially in the earlier layers. By taking an eNTK approximation, TCT optimizes a convex approximation while using information from *all* layers of the model. Empirically, we find that these extracted eNTK features significantly reduce the number of communication rounds needed to learn a performant model, even with data heterogeneity.

## 5 Experiments

We now study the performance of TCT for the decentralized training of deep neural networks in the presence of data heterogeneity. We compare TCT to state-of-the-art federated learning algorithms on three benchmark tasks in federated learning. For each task, we apply these algorithms on client data distributions with varying degrees of data heterogeneity. We find that our proposed approach significantly outperforms existing algorithms when clients have highly heterogeneous data across all tasks. For additional experimental results and implementation details, see Appendix B. Our code is available at `https://github.com/yaodongyu/TCT`.

### 5.1 Experimental Setup

**Datasets and degrees of data heterogeneity.** We assess the performance of federated learning algorithms on the image classification tasks FMNIST [80], CIFAR10, and CIFAR100 [41]. FMNIST and CIFAR10 each consist of 10 classes, while CIFAR100 includes images from 100 classes. There are 60,000 training images in FMNIST, and 50,000 training images in CIFAR10/100.

To vary the degree of data heterogeneity, we follow the setup of Li et al. [45]. We consider two types of non-i.i.d. data distribution: *(i) Data heterogeneity sampled from a symmetric Dirichlet distribution with parameter $\alpha$* [49, 71]. That is, we sample $p_c \sim \mathrm{Dir}_K(\alpha)$ from a $K$-dimensional symmetric Dirichlet distribution and assign a $p_c^k$-fraction of the class $c$ samples to client $k$. (Smaller $\alpha$ corresponds to more heterogeneity.) *(ii) Clients get samples from a fixed subset of classes* [53]. That is, each client is allocated a subset of classes; then, the samples of each class are split into non-overlapping subsets and assigned to clients that were allocated this class. We use #C to denote the number of classes allocated to each client. For example, #C=2 means each client has samples from 2 classes. To allow for consistent comparisons, all of our experiments are run with 10 clients.

**Models.** For FMNIST, we use a convolutional neural network with ReLU activations consisting of two convolutional layers with max pooling followed by two fully connected layers (SimpleCNN). For CIFAR10 and CIFAR100, we mainly consider an 18-layer residual network [25] with 4 basic residual blocks (ResNet-18). In Appendix B.2, we present experimental results for other architectures.

**Algorithms and training schemes.** We compare TCT to state-of-the-art federated learning algorithms, focusing on the widely-used algorithms FedAvg [53], FedProx [48], and SCAFFOLD [39].

Table 2: The top-1 accuracy (%) of our algorithm (TCT) vs. state-of-the-art federated learning algorithms evaluated on FMNIST, CIFAR10, and CIFAR100. We vary the degree of data heterogeneity by controlling the $\alpha$ parameter of the symmetric Dirichlet distribution $\mathrm{Dir}_K(\alpha)$ and the #C parameter for assigning how many labels each client owns. Higher accuracy is better. The highest top-1 accuracy in each setting is highlighted in **bold**.

| Datasets | Architectures | Methods | Non-i.i.d. degree | | | |
|---|---|---|---|---|---|---|
| | | | #C $= 1$ | #C $= 2$ | $\alpha = 0.1$ | $\alpha = 0.5$ |
| FMNIST | SimpleCNN | FedAvg | 35.10% | 85.18% | 86.18% | 90.09% |
| | | FedProx | 50.04% | 84.91% | 86.31% | 89.77% |
| | | SCAFFOLD | 12.80% | 42.80% | 83.87% | 89.40% |
| | | *TCT* | **86.32%** | **90.33%** | **90.78%** | **91.13%** |
| | | *Centralized* | | 91.40% | | |
| | | | #C $= 1$ | #C $= 2$ | $\alpha = 0.1$ | $\alpha = 0.5$ |
| CIFAR-10 | ResNet-18 | FedAvg | 11.27% | 56.86% | 82.60% | 90.43% |
| | | FedProx | 12.30% | 56.87% | 83.31% | 90.68% |
| | | SCAFFOLD | 10.00% | 46.75% | 80.46% | 90.72% |
| | | *TCT* | **49.92%** | **83.02%** | **89.21%** | **91.10%** |
| | | *Centralized* | | 91.90% | | |
| | | | $\alpha = 0.001$ | $\alpha = 0.01$ | $\alpha = 0.1$ | $\alpha = 0.5$ |
| CIFAR-100 | ResNet-18 | FedAvg | 53.89% | 54.22% | 63.49% | 67.65% |
| | | FedProx | 52.87% | 54.32% | 63.47% | 67.54% |
| | | SCAFFOLD | 49.86% | 54.07% | 65.67% | **71.07%** |
| | | *TCT* | **68.42%** | **69.07%** | **69.66%** | 69.68% |
| | | *Centralized* | | 73.61% | | |

(a) FedAvg.  (b) SCAFFOLD.  (c) TCT.

Figure 2: Training/test accuracy vs. communication round for FedAvg (left), SCAFFOLD (middle), and our algorithm TCT (right) on the CIFAR100 dataset with various degrees of non-iid-ness ($\mathrm{Dir}_K(\alpha)$ with $\alpha \in \{0.1, 0.01, 0.001\}$). Dotted lines represent the training accuracy, and dashdot lines with markers represent the test accuracy.

(For comparisons to additional algorithms, see Appendix B.1.) Each client uses SGD with weight decay $10^{-5}$ and batch size 64 by default. For each baseline method, we run it for 200 total communication rounds using 5 local training epochs with local learning rate selected from $\{0.1, 0.01, 0.001\}$ by grid search. For TCT, we run 100 rounds of FedAvg in Stage 1 following the above and use 100 communication rounds in Stage 2 with $M = 500$ local steps and local learning rate $\eta = 5 \cdot 10^{-5}$.

## 5.2 Main Results

Table 2 displays the top-1 accuracy of all algorithm on the three tasks with varying degrees of data heterogeneity. We evaluated each algorithms on each task under four degrees of data heterogeneity. Smaller #C and $\alpha$ in Table 2 correspond to higher heterogeneity.

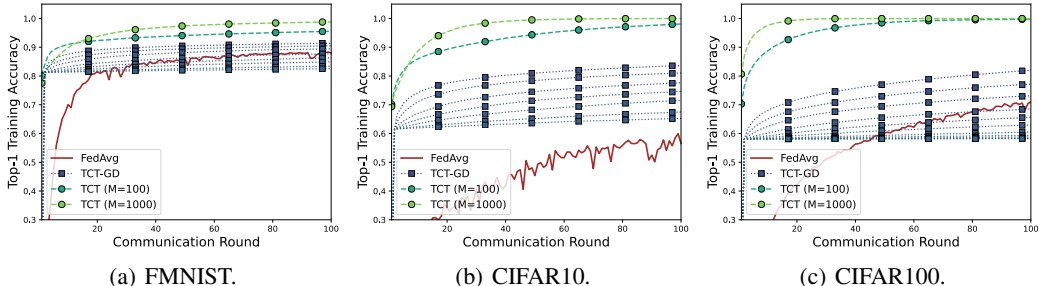

|  (a) FMNIST. | (b) CIFAR10. | (c) CIFAR100. |

Figure 3: Training accuracy vs. communication round for full batch gradient descent (GD) and TCT on FMNIST-[#C=2] **(a)**, CIFAR10-[#C=2] **(b)**, and CIFAR100-[$\alpha = 0.01$] **(c)**. Each dotted line with square markers represents the training accuracy of GD with some learning rate. Dashed lines with circle markers represent the training accuracy of TCT with different numbers of local steps. We also include the training accuracy results of FedAvg with learning rate $\eta = 0.1$. We use TCT-GD to denote the variant of TCT which replaces SCAFFOLD with GD in Stage 2.

We find that the existing federated algorithms all suffer when data heterogeneity is high across all three tasks. For example, the top-1 accuracy of FedAvg on CIFAR-10 is $56.86\%$ when #C=2, which is much worse than the $90.43\%$ achieved in a more homogeneous setting (e.g. $\alpha = 0.5$). In contrast, TCT achieves consistently strong performance, even in the face of high data heterogeneity. More specifically, TCT achieves the best top-1 accuracy performance across all settings except CIFAR-100 with $\alpha = 0.5$, where TCT does only slightly worse than SCAFFOLD.

In absolute terms, we find that TCT is not affected much by data heterogeneity, with performance dropping by less than $1.5\%$ on CIFAR100 as $\alpha$ goes from $0.5$ to $0.001$. Moreover, our algorithm improves over existing methods by at least $15\%$ in the challenging cases, including FMNIST with #C=1, CIFAR-10 with #C=1 and #C=2, and CIFAR-100 with $\alpha = 0.01$ and $\alpha = 0.001$. And, perhaps surprisingly, our algorithm still performs relatively well in the extreme non-i.i.d. setting where each client sees only a single class.

Figure 2 compares the performances of FedAvg, SCAFFOLD, and TCT in more detail on CIFAR100 dataset with different degrees of data heterogeneity. We consider the Dirichlet distribution with parameter $\alpha \in \{0.1, 0.01, 0.001\}$ and compare the training and test accuracy of these three algorithms. As shown in Figures 2(a) and 2(b), both FedAvg and SCAFFOLD struggle when data heterogeneity is high: for both algorithms, test accuracy drops significantly when $\alpha$ decreases. In contrast, we see from Figure 2(c) that TCT maintains almost the same test accuracy for different $\alpha$. Furthermore, the same set of default parameters for our algorithm, including local learning rate and the number of local steps, is relatively robust to different levels of data heterogeneity.

### 5.3 Communication Efficiency

To understand the effectiveness of the local steps in our algorithm, we compare SCAFFOLD (used in TCT-Stage 2) to full batch gradient descent (GD) applied to the overparameterized linear regression problem in Stage 2 of TCT on these datasets. For our algorithm, we set local steps $M \in \{10^2, 10^3\}$ and use the default local learning rate. For full batch GD, we vary the learning rate from $10^{-5}$ to $10^{-1}$ and visualize the ones that do not diverge.

The results are summarized in Figure 3. Each dotted line with square markers in Figure 3 corresponds to full batch GD with some learning rate. Across all three datasets, our proposed algorithm consistently outperforms full batch GD. Meanwhile, we find that more local steps for our algorithms lead to faster convergence across all settings. In particular, our algorithm converges within 20 communication rounds on CIFAR100 (as shown in Figure 3(c)). These results suggest that our proposed algorithm can largely leverage the local computation and improve communication efficiency.

### 5.4 Ablations

**Gradient correction.** We investigate the role of gradient correction when solving overparameterized linear regression with eNTK features in TCT. We compare SCAFFOLD (used in TCT) to FedAvg on solving the regression problems and summarize the results in Figure 4. We use the default

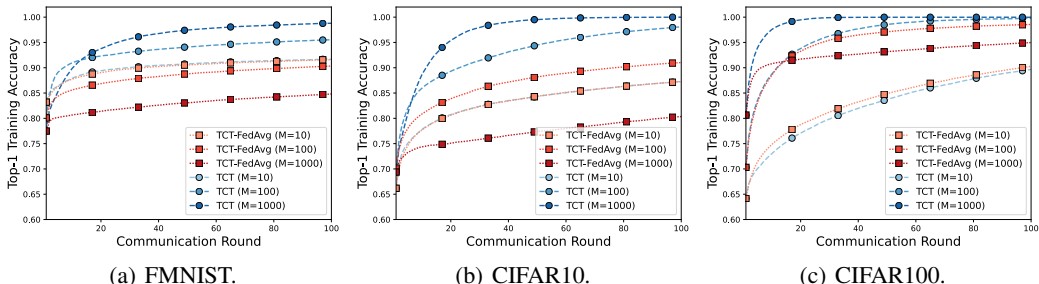

| (a) FMNIST. | (b) CIFAR10. | (c) CIFAR100. |

Figure 4: Comparing TCT to TCT-FedAvg for solving the overparameterized linear regression problem on **(a)** FMNIST-[#C=2], **(b)** CIFAR10-[#C=2], and **(c)** CIFAR100-[$\alpha = 0.01$]. We use TCT-FedAvg to denote a variant of TCT that uses FedAvg instead of SCAFFOLD to perform linear regression in TCT-Stage 2. Dotted red lines with square markers represent the training accuracy of TCT-FedAvg with different numbers of local steps. Dashed blue red lines with circle markers represent the training accuracy of TCT with different numbers of local steps. A darker color means more local steps.

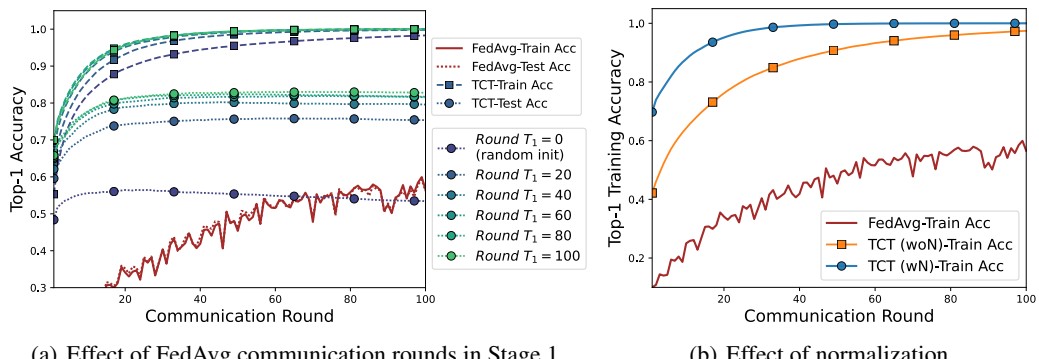

| (a) Effect of FedAvg communication rounds in Stage 1. | (b) Effect of normalization. |

Figure 5: **(a).** We evaluate TCT on using checkpoints save at different communication rounds $T_1$ in Stage 1. $T_1 = 0$ corresponds to the randmon initialized model weights scenario (without FedAvg training). Dash lines with square markers represent the training accuracy, and dotted lines with circle makers represent the test accuracy. **(b).** We study the effect of pre-conditioning on TCT. TCT (wN) corresponds to the setting where eNTK features are normalized, and TCT (woN) corresponds to the without normalization step setting.

local learning rate and consider three different numbers of local steps for both algorithms, i.e., $M \in \{10, 100, 1000\}$. As shown in Figure 4, our approach largely outperforms FedAvg when the number of local steps is large ($M \geq 100$) across three datasets. We also find that the performance of FedAvg can even degrade when the number of local steps increases. For example, FedAvg with $M = 1000$ performs the worst across all three datasets. In contrast to FedAvg, SCAFFOLD converges faster when the number of local steps increases. These observations highlight the importance of gradient correction in our algorithm.

**Model weights for computing eNTK features.** To understand the impact of the model weights trained in Stage 1 of TCT, we evaluate TCT run with different $T_1$ parameters. We consider $T_1 \in \{0, 20, 40, 60, 80, 100\}$, where $T_1 = 0$ corresponds to randomly initialized weights. From Figure 5(a), we find that weights after FedAvg training are much more effective than weights at random initialization. Specifically, without FedAvg training, the eNTK (at random initialization) performs worse than standard FedAvg. In contrast, TCT significantly outperforms FedAvg by a large margin (roughly $20\%$ in test accuracy) when eNTK features are extracted from a FedAvg-trained model. Also, we find that TCT is stable with respect to the choice of communication rounds $T_1$ in Stage 1. For example, models trained by TCT with $T_1 \geq 60$ achieve similar performance.

**Effect of normalization.** In Figure 5(b), we investigate the role of normalization on TCT by comparing TCT run with normalized and unnormalized eNTK features. The same number of local

steps ($M = 500$) is applied for both settings. We tune the learning rate $\eta$ for each setting and plot the run that performs best (as measured in training accuracy). The results in Figure 5(b) suggest that the normalization step in TCT significantly improves the communication efficiency by increasing convergence speed. In particular, TCT with normalization converges to nearly $100\%$ training accuracy in approximately 40 communication rounds, which is much faster than TCT without normalization.

**Pre-training vs. Bootstrapping.** In Appendix B.4, we explore the effect of starting from a pre-trained model instead of relying on bootstrapping to learn the features. We find that pre-training further improves the performance of TCT and completely erases the gap between centralized and federated learning.

Additionally, we conduct experiments on investigating the role of training loss function and subsampling approximation in TCT-Stage 2. For TCT-Stage 2, we find that neither using the cross-entropy loss as the training objective nor applying full eNTK representations significantly improves the performance of TCT. On the other hand, applying subsampling approximation in TCT-Stage 2 can largely improve the communication efficiency compared to the full eNTK representations approach. See Appendix B.7 for detailed experimental results.

## 6 Conclusion

We have argued that nonconvexity poses a significant challenge for federated learning algorithms. We found that a neural network trained in such a manner does learn useful features, but fails to use them and thus has poor overall accuracy. To sidestep this issue, we proposed a *train-convexify-train* procedure: first, train the neural network using FedAvg; then, optimize (using SCAFFOLD) a convex approximation of the model obtained using its empirical neural tangent kernel. We showed that the first stage extracts meaningful features, whereas the second stage learns to utilize these features to obtain a highly performant model. The resulting algorithm is significantly faster and more stable to hyper-parameters than previous federated learning methods. Finally, we also showed that given a good pre-pretrained feature extractor, our convexify-train procedure fully closes the gap between centralized and federated learning.

Our algorithm adds to the growing body of work using eNTK to *linearize* neural networks and obtain tractable convex approximations. However, unlike most of these past works which only work with pre-trained models, our bootstrapping allows training models from scratch. Finally, we stress that the success of our approach underscores the need to revisit theoretical understanding of heterogeneous federated learning. Nonconvexity seems to play an outsized role but its effect in FL has hitherto been unexplored. In particular, black-box notions of difficulty such as gradient dissimilarity or distances between client optima seem insufficient to capture practical performance. It is likely that further progress in the field (e.g. federated pre-training of foundational models), will require tackling the issue of nonconvexity head on.

## Acknowledgments and Disclosure of Funding

We would like to thank the anonymous reviewers for their constructive suggestions and comments. Yaodong Yu acknowledges support from the joint Simons Foundation-NSF DMS grant #2031899. Alexander Wei acknowledges support from an NSF Graduate Research Fellowship under grant DGE2146752. Sai Praneeth Karimireddy acknowledges support of an SNSF postdoc mobility fellowship. Yi Ma acknowledges support from ONR grants N00014-20-1-2002 and N00014-22-1-2102 and the joint Simons Foundation-NSF DMS grant #2031899. Michael Jordan acknowledges support of the ONR Mathematical Data Science program.

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
