# Appendix

## A   Additional Details About Our Algorithm

### A.1   An Efficient Implementation of SCAFFOLD

---

**Algorithm 1** Efficient implementation of SCAFFOLD

---

**Input:** losses $\{L_k\}$, $k \in [K]$. Number of local steps $M$, server model $\theta^0$, learning rate $\eta$.
**Initialization:**    client corrections $\{h_k^{-1} = \mathbf{0}\}$, local models$\{\widehat{\theta}_i^0\} = \theta^0$, $k \in [K]$
**for** round $t = 0, 1, \ldots, T$ **do**
   **for** clients $k = 1, \ldots, K$ in parallel **do**
     Receive $\theta^t$ from server. Update correction

$$h_k^t = h_k^{t-1} + \frac{1}{M\eta}(\theta^t - \widehat{\theta}_k^t). \tag{2}$$

     Initialize client local model $\widehat{\theta}_i^{t,0} = \theta^t$.
     **for** $m = 1, \ldots, M$ **do**
       Update with a stochastic gradient sampled from local client data

$$\widehat{\theta}_k^{t,m+1} = \widehat{\theta}_k^{t,m} - \eta \left( \nabla L_k(\theta_i^{t,m}; \xi_k^{t,m}) - h_k^t \right). \tag{3}$$

     **end for**
     Set $\widehat{\theta}_k^{t+1} = \widehat{\theta}_k^{t,M+1}$. Communicate $\widehat{\theta}_k^{t+1}$ to server.
   **end for**
   Aggregate $\theta^{t+1} = \frac{1}{K}\sum_{k=1}^K \widehat{\theta}_k^{t+1}$.[5]
**end for**

---

We describe a more communication efficient implementation of SCAFFOLD which is equivalent to Option II of SCAFFOLD from [39]. Our implementation only requires a single model to be communicated between the client and server each round, making its communication complexity exactly equivalent to that of FedAvg. To see the equivalence, we prove that our implementation satisfies the following condition for any time step $t \geq 0$:

$$c_k^{t+1} := \frac{1}{M} \sum_{m \in [M]} \nabla L_k(\theta_k^{t,m}; \xi_k^{t,m}), \text{ and}$$

$$c^{t+1} := \frac{1}{K} \sum_{k \in [K]} c_k^t, \text{ we maintain the invariant that}$$

$$h_k^{t+1} = c_k^{t+1} - c^{t+1}.$$

To see this, note that the local client model after updating in round $t$ is

$$
\begin{aligned}
\widehat{\theta}_k^{t+1} &= \widehat{\theta}_k^{t,M+1} \\
&= \theta^t - \eta \sum_{k \in [K]} \nabla L_k(\theta_k^{t,m}; \xi_k^{t,m}) - h_k^t \\
&= \theta^t - M\eta(c_k^{t+1} - h_k^t).
\end{aligned}
$$

By averaging this over the clients, we can see that the server model is

$$\theta^{t+1} = \theta^t - M\eta\Big(c^{t+1} - \frac{1}{K}\sum_{l \in [K]} h_l^t\Big).$$

---

[5]Note that when different clients have different number of data points, the actual aggregation step is $\theta^{t+1} = \sum_{k=1}^K \left( n_k / \sum_j n_j \right) \widehat{\theta}_k^{t+1}$. However, we present the simplified version with equal weights for all clients to ease the comparison with the pseudocode in Karimireddy et al. [39].

By induction, suppose that $h_k^t = c_k^t - c^t$. This implies that summing over the clients, it becomes zero; i.e., $\sum_{l \in [K]} h_l^t = 0$. Plugging this and the previous computations, we have

$$
\begin{aligned}
h_k^{t+1} &= h_k^t + \tfrac{1}{M\eta}(\theta^{t+1} - \widehat{\theta}_k^{t+1}) \\
&= h_k^t + \tfrac{1}{M\eta}(-M\eta c^{t+1} + M\eta(c_k^{t+1} - h_k^t)) \\
&= c_k^{t+1} - c^{t+1} \, .
\end{aligned}
$$

For the base step at $t = 0$, note that $h_i^0 = \mathbf{0}$. This completes the proof by induction.

## A.2 Additional Implementation Details

---

**Algorithm 2** TCT: complete pseudo-code

---

**Input:** input dim $D$, output dim $C$, loss $\ell(\cdot, \cdot) : \mathbb{R}^{C \times C} \to \mathbb{R}$, aggr. weights $\{w_1, \ldots, w_K\}$, model $f$ with parameters $\theta \in \mathbb{R}^P$: $f(x; \theta) = \phi \circ \omega (x) : \mathbb{R}^D \to \mathbb{R}^C$ (e.g., ResNet18), composed of a feature extractor $\phi : \mathbb{R}^D \to \mathbb{R}^E$ and final linear layer $\omega : \mathbb{R}^E \to \mathbb{R}^C$.
**Hyper-parameters:** Local steps $M$ (default 500), Stage-1 lr $\eta_1$ (default 0.01), Stage-1 rounds $T_1$ (default 100), Stage-2 lr $\eta_2$ (default $5 \cdot 10^{-5}$), Stage-2 rounds $T_2$ (default 100).

**Stage 1 (Bootstrapping):**
Initialize server model $\theta^0$.
**for** round $t = 0, 1, \ldots, T_1$ **do**
    **for** clients $k = 1, \ldots, K$ in parallel **do**
        Receive $\theta^t$ from server and initialize client local model $\widehat{\theta}_k^{t,0} = \theta^t$.
        **for** $m = 1, \ldots, M$ **do**
            Update with a mini-batch gradient sampled from local client data $(x_k^{t,m}, y_k^{t,m})$

$$
\widehat{\theta}_k^{t,m+1} = \widehat{\theta}_k^{t,m} - \eta_i \left( \nabla_\theta \ell(f(x_k^{t,m}; \theta_k^{t,m}), y_k^{t,m}) - h_k^t \right).
$$

        **end for**
        Communicate $\widehat{\theta}_k^{t+1}$ to server.
    **end for**
    Aggregate $\theta^{t+1} = \frac{1}{\sum_k w_k} \sum_{k=1}^K w_k \widehat{\theta}_k^{t+1}$ .
**end for**

**Stage 2 (Convexification):**
Input: Bootstrapped parameters $\theta^B$ decomposed as $\theta^B = \phi^B \circ \omega^B$.
Randomly re-initialize using fixed seed linear layer $\omega^r$ and define $\theta^0 := \phi^B \circ \omega^r$.
[Comment:] *Define basis vector* $\mathbf{e_1} := (1, 0, \ldots, 0)$. *For input $x$, $(\mathbf{e}_1^\top f(x; \theta^0))$ is the first logit.*
Optionally, compute a random sub-sampling mask $\mathcal{S}(\phi)$ over feature params using fixed seed .
[Comment:] *For a given input $x$, we will learn parameters $(\varphi, b)$ for prediction as*

$$
\hat{y} = \varphi^\top \phi_{\text{eNTK}}(x) + b, \quad \text{where} \quad \phi_{\text{eNTK}}(x) := \mathcal{S} \left( \nabla_\phi (\mathbf{e}_1^\top f(x; \phi^B \circ \omega^r)) \right).
$$

Compute normalized eNTK features $\tilde{\phi}_{\text{eNTK}}(x)$ (mean 0 and variance 1) across clients.
Also normalize targets to mean 0 using $\tilde{y} := y - \frac{1}{C} \mathbf{1}$.
Run SCAFFOLD (Algorithm 1) over params $\psi := (\varphi, b)$ with learning rate $\eta_2$, local steps $M$, initial server params: $\psi^0 = \mathbf{0}$, and client losses $\{L_k\}$ defined over the local data as

$$
L_k(\psi) := \sum_{(x_k, y_k)} \left( \varphi^\top \tilde{\phi}_{\text{eNTK}}(x_k) + b - \tilde{y}_k \right)^2 .
$$

---

**Additional details about linear regression in TCT.** In our experiments, we normalize the one-hot encoded label of each sample so that the normalized one-hot encoded label has mean 0. More specifically, we subtract $[1/C, \ldots, 1/C]^\top \in \mathbb{R}^{C \times 1}$ from the one-hot encoding label vector, where $C$ is the number of classes. Further, Hui and Belkin [30] show that performance for large number

**Algorithm 3** Compute eNTK Pseudocode, PyTorch-like

```python
def compute_eNTK(model, X, num_params, subsample_size=100000, seed=123):
    """compute eNTK of input X with model"""
    # model: model for linearization
    # X: (n x d), n -- number of samples, d -- input dimension
    # subsample_size: parameter of subsampling operation
    # seed: random seed for subsampling operation
    # num_params: total number of parameters for model
    model.eval()
    params = list(model.parameters())
    torch.manual_seed(seed)
    torch.cuda.manual_seed(seed)
    random_index = torch.randperm(num_params)[:subsample_size]
    eNTKs = torch.zeros((X.size()[0], subsample_size))
    for i in range(X.size()[0]):
        # compute eNTK for the i-th input
        model.zero_grad()
        model.forward(X[i:i+1])[0].backward()
        eNTK = []
        for param in params:
            if param.requires_grad:
                eNTK.append(param.grad.flatten())
        eNTK = torch.cat(eNTK)
        # subsampling
        eNTKs[i, :] = eNTK[random_index]
    return eNTKs
```

of classes can be improved by increasing the penalty for mis-classification and scaling the target from 1 to a larger value (e.g., 30). Achille et al. [2] show that using Leaky-ReLU, and using K-FAC preconditioning further improves the performance. However, we do not explore such optimizations in this work–these (and other optimization tricks for least-squares regression) can be easily incorporated into our framework.

**Local learning rate for TCT.** From our experiments, we find that small local learning rates ($\eta \leq 10^{-4}$) achieve good train/test accuracy performance for TCT with the normalization step. When the normalization step in TCT is applied, larger local learning rates diverge. Meanwhile, local learning rates from $[10^{-6}, 10^{-4}]$ achieve similar performance for TCT (as shown in Table 8). On the other hand, without the normalization step, TCT with large learning rate ($\eta \in [0.01, 0.5]$) does not diverge. When running more communication rounds, TCT (without the normalization step) with large learning rate achieves similar performance as the default TCT (with the normalization step).

**Additional details about Stage 2 of TCT.** To solve the linear regression problem in TCT-**Stage 2**, we use the full batch gradient in Eq. (3) of Algorithm 1 in our implementation.

**Additional details about Figure 5.** We consider CIFAR10-[#C=2] in Figure 5(a) and 5(b).

**Details about the total amount of compute.** We use NVIDIA 2080 Ti, A4000, and A100 GPUs, and our experiments required around 500 hours of GPU time.

# B  Additional Experimental Results

## B.1  Additional Baselines

**In comparison with FedAdam and FedDyn.** We compare TCT to FedAdam [61], FedDyn [1], and FedNova [71] in Table 3. We consider four settings in Table 3, including CIFAR10 (#C $= 2$), CIFAR10 ($\alpha = 0.1$), CIFAR100 ($\alpha = 0.001$), and CIFAR100 ($\alpha = 0.01$). For FedDyn, we perform similar hyperparameter selection as FedAvg; i.e., select local learning rate from $\{0.1, 0.01, 0.001\}$. For FedAdam, following recommendation by [61], we set the global learning rate as $\eta_{\text{global}} = 0.1$ and select local learning rate from $\{10^{-1}, 10^{-1.5}, 10^{-2}, 10^{-2.5}, 10^{-3}\}$. Similar to results in Table 2, we find that TCT significantly outperforms the existing methods in high data heterogeneity settings.

Table 3: The top-1 test accuracy (%) of our algorithm (TCT) vs. other federated learning algorithms (FedAdam [61], FedDyn [1], and FedNova [71]) evaluated on CIFAR10 and CIFAR100. We vary the degree of data heterogeneity by controlling the $\alpha$ parameter of the symmetric Dirichlet distribution $\text{Dir}_K(\alpha)$ and the #C parameter for assigning how many labels each client owns. Higher accuracy is better. The highest top-1 accuracy in each setting is highlighted in **bold**.

| Methods | Datasets | | | |
|---|---|---|---|---|
| | CIFAR10 (#C $= 2$) | CIFAR10 ($\alpha = 0.1$) | CIFAR100 ($\alpha = 0.001$) | CIFAR100 ($\alpha = 0.01$) |
| FedAdam [61] | 33.52% | 62.57% | 30.85% | 37.16% |
| FedDyn [1] | 51.67% | 81.03% | 50.86% | 53.79% |
| FedNova [71] | 53.27% | 84.26% | 56.06% | 58.47% |
| TCT | **83.02%** | **89.21%** | **69.07%** | **69.66%** |

## B.2  Results of Other Architectures

In Section 5, we use batch normalization [32] as the default normalization layer on CIFAR10 and CIFAR100 datasets, and we denote the ResNet-18 with batch normalization layers by ResNet-18-BN. In Table 4, we consider group normalization [79] on CIFAR10 and CIFAR100 and let ResNet-18-GN denote the ResNet-18 with group normalization. We set `num_groups=2` in group normalization layers. As shown in Table 4, TCT achieves better performance than FedAvg with ResNet-18-GN on both CIFAR10 and CIFAR100 datasets. Our experiments indicate that in extremely heterogeneous settings, group norm is insufficient to fix FedAvg.

Table 4: The top-1 test accuracy (%) of our algorithm (TCT) vs. FedAvg(-GN) evaluated on CIFAR10 and CIFAR100. We vary the degree of data heterogeneity by controlling the $\alpha$ parameter of the symmetric Dirichlet distribution $\text{Dir}_K(\alpha)$ and the #C parameter for assigning how many labels each client owns. Higher accuracy is better. The highest top-1 accuracy in each setting is highlighted in **bold**.

| Datasets | Architectures | Methods | Non-i.i.d. degree | | | |
|---|---|---|---|---|---|---|
| | | | #C $= 1$ | #C $= 2$ | $\alpha = 0.1$ | $\alpha = 0.5$ |
| | ResNet-18-GN | FedAvg | 21.23% | 56.80% | 84.72% | 89.03% |
| CIFAR-10 | ResNet-18-BN | FedAvg | 11.27% | 56.86% | 82.60% | 90.43% |
| | ResNet-18-BN | *TCT* | **49.92%** | **83.02%** | **89.21%** | **91.10%** |
| | | | $\alpha = 0.001$ | $\alpha = 0.01$ | $\alpha = 0.1$ | $\alpha = 0.5$ |
| | ResNet-18-GN | FedAvg | 47.60% | 48.60% | 53.29% | 55.39% |
| CIFAR-100 | ResNet-18-BN | FedAvg | 53.89% | 54.22% | 63.49% | 67.65% |
| | ResNet-18-BN | *TCT* | **68.42%** | **69.07%** | **69.66%** | **69.68%** |

## B.3 Additional Experimental Results of the Effect of Stage 1 Communication Round for TCT

In Figure 6, we provide additional results of the effect of $T_1$ for TCT on CIFAR10 and CIFAR100 datasets. We find that TCT outperforms existing algorithm across all $T_1$ communication rounds, where $T_1 \geq 20$. Extending the number of rounds for the baseline algorithms to 200 rounds does not improve their performance. In contrast, running 60 rounds of bootstrapping using FedAvg followed by 40 rounds of TCT gives near-optimal performance across all settings.

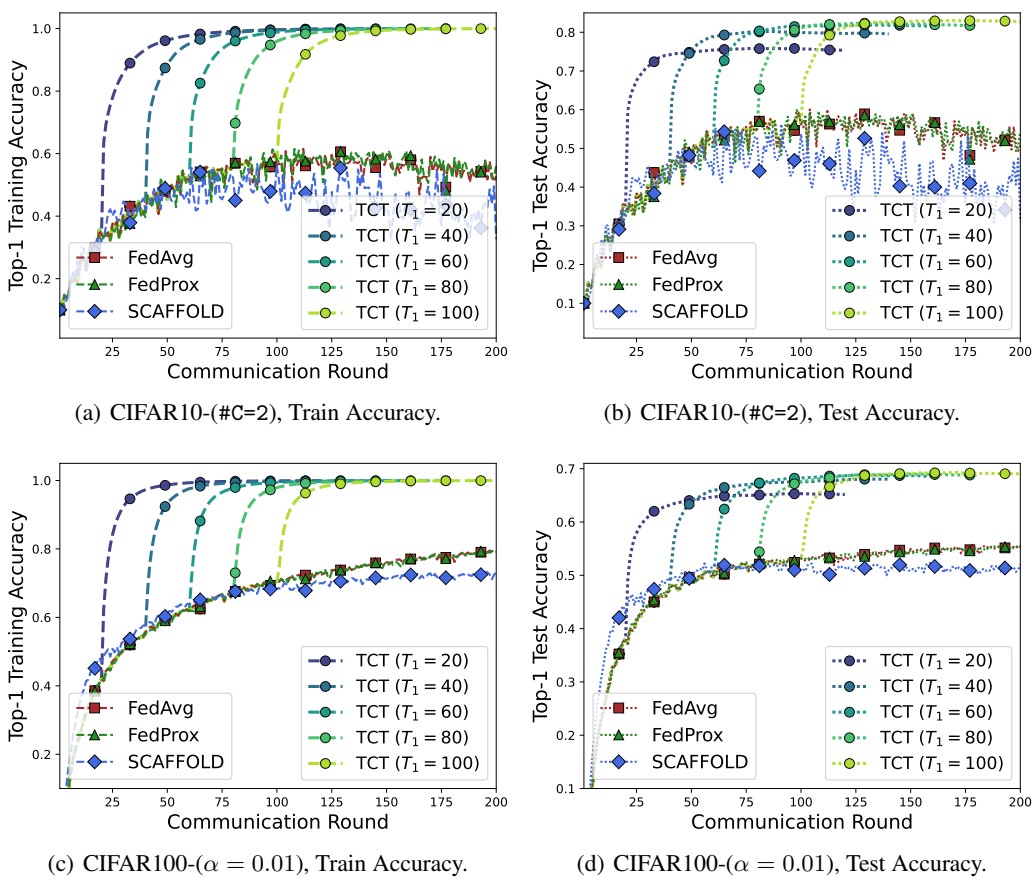

(a) CIFAR10-(#C=2), Train Accuracy.

(b) CIFAR10-(#C=2), Test Accuracy.

(c) CIFAR100-($\alpha = 0.01$), Train Accuracy.

(d) CIFAR100-($\alpha = 0.01$), Test Accuracy.

Figure 6: We evaluate TCT on using checkpoints saved at different communication rounds $T_1$ in **Stage 1**. We compare TCT to existing algorithm, including FedAvg, FedProx, and SCAFFOLD. For all three existing algorithms, we visualize the results of local learning rate $\eta = 0.1$. The train/test accuracy results in the first $T_1$ communication rounds of TCT are the same as FedAvg. For example, "TCT ($T_1 = 20$)" corresponds to training the model with FedAvg for $T_1 = 20$ rounds in **Stage 1** and then running 100 rounds of SCAFFOLD for solving the linear regression problem in **Stage 2**. Plots **(a)** and **(c)** display training accuracy and **(b)** and **(d)** display test accuracy.

## B.4 Additional Experimental Results of Pre-trained Models

In Table 5 and Figure 7, we provide additional results of the effect of pre-training for FedAvg and TCT on CIFAR10 and CIFAR100 datasets. For both methods, we use the ResNet-18 pre-trained on ImageNet-1k [25] as the initialization. We use *FedAvg (last layer)* to denote applying FedAvg on learning the last linear layer of the model, i.e., layers except for the last linear layer are frozen during training. Compared to results in Table 2, we find that using pre-trained model as initialization largely improves the performance of both FedAvg and TCT. However, FedAvg still suffers from data heterogeneity. In contrast, TCT achieves similar performance as the centralized setting on both datasets across different degrees of data heterogeneity.

Table 5: The top-1 test accuracy (%) of our algorithm (TCT) vs. FedAvg evaluated on CIFAR10 and CIFAR100 with pre-trained model initialization. We vary the degree of data heterogeneity by controlling the $\alpha$ parameter of the symmetric Dirichlet distribution $\mathrm{Dir}_K(\alpha)$ and the #C parameter for assigning how many labels each client owns. Higher accuracy is better. The highest top-1 accuracy in each setting is highlighted in **bold**.

| Methods | Datasets | | | |
|---|---|---|---|---|
| | CIFAR10 (#C = 2) | CIFAR10 ($\alpha = 0.1$) | CIFAR100 ($\alpha = 0.001$) | CIFAR100 ($\alpha = 0.01$) |
| Centralized | 95.13% | 95.13% | 80.65% | 80.65% |
| FedAvg (last layer) | 63.60% | 75.16% | 50.40% | 51.97% |
| FedAvg | 64.73% | 84.25% | 62.23% | 63.81% |
| TCT | **92.97%** | **93.70%** | **79.25%** | **79.55%** |

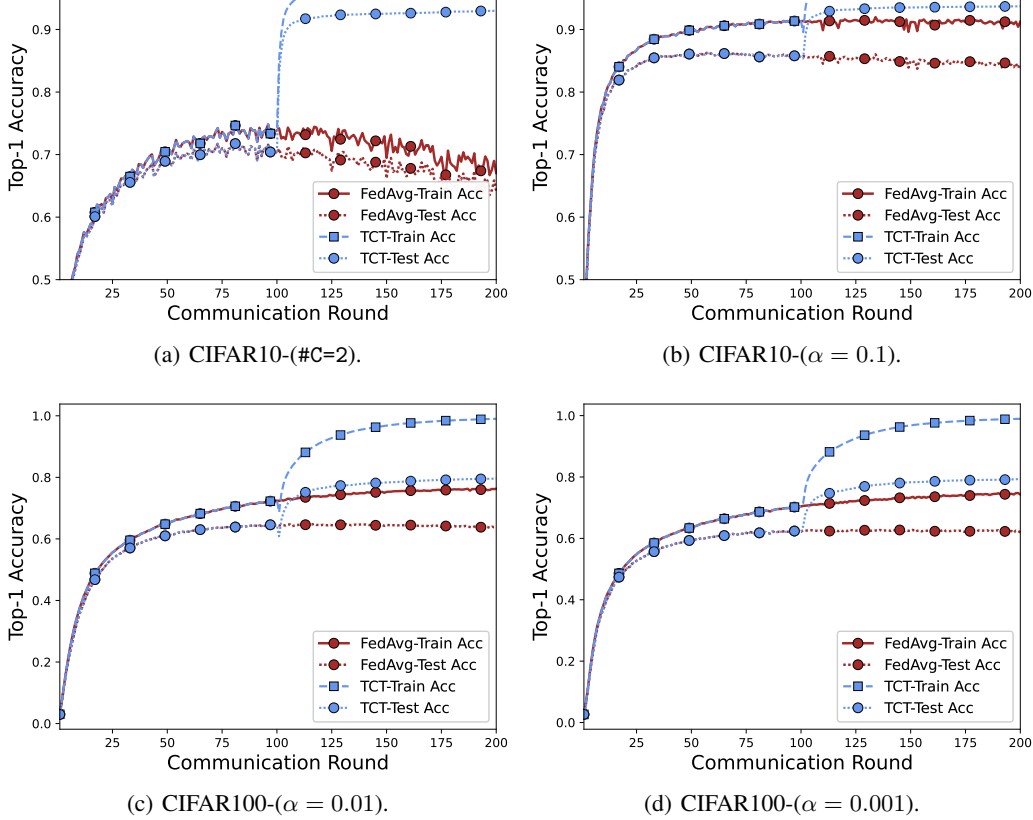

(a) CIFAR10-(#C=2).

(b) CIFAR10-($\alpha = 0.1$).

(c) CIFAR100-($\alpha = 0.01$).

(d) CIFAR100-($\alpha = 0.001$).

Figure 7: We evaluate FedAvg and TCT on CIFAR10 and CIFAR100 datasets with pre-trained ResNet-18. Plots **(a)** and **(b)** display training/test accuracy on the CIFAR10 dataset and **(c)** and **(d)** display training/test accuracy on the CIFAR100 dataset.

## B.5 Additional Experimental Results of One-round Communication

In Table 6, we provide additional results of TCT on CIFAR10 and CIFAR100 datasets with one communication round in TCT-Stage 2. Specifically, we set the number of local steps $M = 500$, local learning rate $\eta = 0.00005$, and the total number of communication round $T = 1$ in TCT-Stage 2. The results are summarized in Table 6. With only one communication round in TCT-Stage 2, TCT still achieves better performance than FedAvg in three out of four settings in Table 6. On the other hand, we recommend setting the communication round in TCT-Stage 2 larger than 10 for our method TCT in order to achieve satisfying performance.

Table 6: The top-1 test accuracy (%) of our algorithm (TCT) on CIFAR10 and CIFAR100 with one communication round in TCT-Stage 2.

| Methods | Datasets | | | |
|---|---|---|---|---|
| | CIFAR10 (#C $= 2$) | CIFAR10 ($\alpha = 0.01$) | CIFAR100 ($\alpha = 0.001$) | CIFAR100 ($\alpha = 0.01$) |
| FedAvg | 56.86% | 82.60% | 53.89% | 54.22% |
| TCT | 83.02% | 89.21% | 68.42% | 69.07% |
| TCT-OneRound | 64.94% | 82.62% | 55.50% | 57.51% |

## B.6 Additional Experimental Results of Large Number of Clients

We study the performance of our proposed algorithm as well as existing algorithms in the large number of clients setting, where we consider the number of clients $K = 50$ on CIFAR100 with $\alpha = 0.001$. The results are summarized in Table 7. We find our proposed method (TCT: 45.32%) significantly outperforms existing methods (best test accuracy: 16.70%).

Table 7: The top-1 accuracy (%) of our algorithm (TCT) vs. state-of-the-art federated learning algorithms evaluated on CIFAR100 with a large number of clients. We set the degree of data heterogeneity parameter $\alpha = 0.001$ and set the total number of clients $K = 50$. Higher accuracy is better. The highest top-1 accuracy is highlighted in **bold**.

| Dataset | Architecture | # Clients | FedAvg | FedProx | SCAFFOLD | TCT |
|---|---|---|---|---|---|---|
| CIFAR100 ($\alpha = 0.001$) | ResNet-18 | 50 | 16.70% | 16.24% | 13.41% | **45.32%** |

## B.7 Additional Ablations

**Effect of local learning rate for TCT and FedAdam.** As mentioned in Reddi et al. [61], FedAdam is more robust to the choice of local learning rate compared to FedAvg. We conduct additional ablations on the effect of local learning rate for TCT as well as FedAdam [61] on the CIFAR10 dataset. For each algorithm, we first select a base local learning rate $\eta_{\text{base}}$ and then vary the local learning rate $\eta \in \{\eta_{\text{base}} \cdot 10^0, \eta_{\text{base}} \cdot 10^{-0.5}, \eta_{\text{base}} \cdot 10^{-1.0}, \eta_{\text{base}} \cdot 10^{-1.5}\}$. The results are summarized in Table 8. Compared to FedAdam, we find that TCT is much less sensitive to the choice of local learning rate.

Table 8: The top-1 test accuracy (%) of our algorithm (TCT) and FedAdam [61] evaluated on the CIFAR10 dataset. We vary the local learning rate for both algorithms. Higher accuracy is better.

| Datasets | Methods | Local learning rate | | | |
|---|---|---|---|---|---|
| | ($\eta_{\text{base}} = 10^{-4}$) | $\eta = \eta_{\text{base}} \cdot 10^0$ | $\eta = \eta_{\text{base}} \cdot 10^{-0.5}$ | $\eta = \eta_{\text{base}} \cdot 10^{-1.0}$ | $\eta = \eta_{\text{base}} \cdot 10^{-1.5}$ |
| CIFAR10-(#C=2) | TCT | 82.12% | 83.60% | 83.51% | 82.37% |
| CIFAR10-($\alpha = 0.1$) | TCT | 88.68% | 89.27% | 89.23% | 89.15% |
| | ($\eta_{\text{base}} = 10^{-1.5}$) | $\eta = \eta_{\text{base}} \cdot 10^0$ | $\eta = \eta_{\text{base}} \cdot 10^{-0.5}$ | $\eta = \eta_{\text{base}} \cdot 10^{-1.0}$ | $\eta = \eta_{\text{base}} \cdot 10^{-1.5}$ |
| CIFAR10-(#C=2) | FedAdam [61] | 31.29% | 33.52% | 26.20% | 14.96% |
| CIFAR10-($\alpha = 0.1$) | FedAdam [61] | 10.31% | 37.26% | 62.57% | 49.18% |

**Effect of local learning rate and number of local steps for TCT.**   We conduct additional ablations on the effect of both the local learning rate $\eta$ and the number of local steps $M$ for TCT on CIFAR10 and CIFAR100 datasets. The results in Table 9 and Table 10 indicate that TCT is robust to the choice of the local learning rate $\eta$ and the number of local steps $M$. We find that as the number of steps increases, the learning rate should predictably decrease. The performance is relatively stable along the diagonal, indicating that it is the product $M \cdot \eta$ which affects accuracy.

Table 9: The top-1 test accuracy (%) of our algorithm (TCT) evaluated on the CIFAR10 dataset. We consider CIFAR10-(#C=2) and we set $\eta_{\text{base}} = 10^{-4}$. We vary both the local learning rate and the number of local steps for TCT. Higher accuracy is better.

| Number of local steps | Local learning rate | | | |
|---|---|---|---|---|
| | $\eta = \eta_{\text{base}} \cdot 10^{0}$ | $\eta = \eta_{\text{base}} \cdot 10^{-0.5}$ | $\eta = \eta_{\text{base}} \cdot 10^{-1.0}$ | $\eta = \eta_{\text{base}} \cdot 10^{-1.5}$ |
| $M = 50$ | 83.51% | 82.37% | 80.67% | 78.14% |
| $M = 100$ | 83.59% | 83.35% | 81.73% | 79.71% |
| $M = 500$ | 82.12% | 83.60% | 83.51% | 82.37% |
| $M = 1000$ | 80.78% | 82.92% | 83.59% | 83.35% |

Table 10: The top-1 test accuracy (%) of our algorithm (TCT) evaluated on the CIFAR100 dataset. We consider CIFAR100-($\alpha = 0.01$) and we set $\eta_{\text{base}} = 10^{-4}$. We vary both the local learning rate and the number of local steps for TCT. Higher accuracy is better.

| Number of local steps | Local learning rate | | | |
|---|---|---|---|---|
| | $\eta = \eta_{\text{base}} \cdot 10^{0}$ | $\eta = \eta_{\text{base}} \cdot 10^{-0.5}$ | $\eta = \eta_{\text{base}} \cdot 10^{-1.0}$ | $\eta = \eta_{\text{base}} \cdot 10^{-1.5}$ |
| $M = 50$ | 69.12% | 67.31% | 64.61% | 61.34% |
| $M = 100$ | 69.54% | 68.48% | 66.43% | 63.42% |
| $M = 500$ | 69.03% | 69.60% | 69.12% | 67.31% |
| $M = 1000$ | 68.38% | 69.42% | 69.54% | 68.48% |

**Effect of training loss for TCT-Stage 2.**   We compare the performance of quadratic loss (defined in Eq. (1)) and cross-entropy loss (i.e., applying the cross-entropy loss for learning the linear model in TCT-Stage 2, denoted by *TCT-CE*) for TCT in Table 11. As shown in Table 11, we find quadratic loss indeed achieves better performance than cross-entropy loss for TCT.

**Effect of subsampling for TCT-Stage 2.**   We study the performance of full eNTK representation in TCT-Stage 2 (i.e., without random subsampling) to investigate the role of the subsampling approximation. We provide the results in Table 11. As shown in Table 11, applying full eNTK representations slightly improves (improvements are smaller than 2% across all settings) the performance of TCT on CIFAR10/100. On the other hand, using subsampled eNTK reduces the communication cost more than 100x compared to the full eNTK and existing federated learning algorithms (#parameter of the whole model: 11,169,345, #parameters of the subsample eNTK: 100,000).

**Applying last layer representations in TCT-Stage 2.**   We study the performance of only applying last layer representations in TCT-Stage 2, and the results are summarized in Table 12. From Table 12, we find that applying eNTK representations with high dimension (i.e., $p = 100,000$) outperforms using the representations before the last layer only, especially in the settings with high degrees of data heterogeneity. These results provide further evidence on applying eNTK features instead of the representations before the last layer features.

Table 11: The top-1 accuracy (%) of our algorithm (TCT) vs. TCT-CE, TCT-full-eNTK on FMNIST, CIFAR10, and CIFAR100. We vary the degree of data heterogeneity by controlling the $\alpha$ parameter of the symmetric Dirichlet distribution $\mathrm{Dir}_K(\alpha)$ and the #C parameter for assigning how many labels each client owns. Higher accuracy is better. TCT-CE represents the variant of TCT where we apply cross-entropy loss in Stage 2 of TCT. TCT-full-eNTK represents the variant of TCT where we use the full eNTK representation (without subsampling) in Stage 1 of TCT.

| Datasets | Architectures | Methods | Non-i.i.d. degree | | | |
|---|---|---|---|---|---|---|
| | | | #C $= 1$ | #C $= 2$ | $\alpha = 0.1$ | $\alpha = 0.5$ |
| | | *TCT* | 86.32% | 90.33% | 90.78% | 91.13% |
| FMNIST | SimpleCNN | *TCT-CE* | 86.50% | 89.23% | 89.66% | 90.15% |
| | | *TCT-full-eNTK* | 86.32% | 90.36% | 90.90% | 91.18% |
| | | | #C $= 1$ | #C $= 2$ | $\alpha = 0.1$ | $\alpha = 0.5$ |
| | | *TCT* | 49.92% | 83.02% | 89.21% | 91.10% |
| CIFAR-10 | ResNet-18 | *TCT-CE* | 45.13% | 81.06% | 88.03% | 91.12% |
| | | *TCT-full-eNTK* | 50.38% | 84.92% | 89.72% | 91.69% |
| | | | $\alpha = 0.001$ | $\alpha = 0.01$ | $\alpha = 0.1$ | $\alpha = 0.5$ |
| | | *TCT* | 68.42% | 69.07% | 69.66% | 69.68% |
| CIFAR-100 | ResNet-18 | *TCT-CE* | 63.46% | 64.08% | 65.22% | 66.07% |
| | | *TCT-full-eNTK* | 69.81% | 70.05% | 70.12% | 70.91% |

Table 12: The top-1 accuracy (%) of our algorithm (TCT) vs. TCT-last-layer on FMNIST, CIFAR10, and CIFAR100. We vary the degree of data heterogeneity by controlling the $\alpha$ parameter of the symmetric Dirichlet distribution $\mathrm{Dir}_K(\alpha)$ and the #C parameter for assigning how many labels each client owns. Higher accuracy is better. The highest top-1 accuracy in each setting is highlighted in **bold**. TCT-last-layer represents the variant of TCT where we apply the representations of last layer and cross-entropy loss in Stage 2 of TCT.

| Datasets | Architectures | Methods | Non-i.i.d. degree | | | |
|---|---|---|---|---|---|---|
| | | | #C $= 1$ | #C $= 2$ | $\alpha = 0.1$ | $\alpha = 0.5$ |
| FMNIST | SimpleCNN | *TCT* | **86.32%** | **90.33%** | **90.78%** | **91.13%** |
| | | *TCT-last-layer* | 60.43% | 83.96% | 86.01% | 89.33% |
| | | | #C $= 1$ | #C $= 2$ | $\alpha = 0.1$ | $\alpha = 0.5$ |
| CIFAR-10 | ResNet-18 | *TCT* | **49.92%** | **83.02%** | **89.21%** | **91.10%** |
| | | *TCT-last-layer* | 35.51% | 74.55% | 86.57% | 90.76% |
| | | | $\alpha = 0.001$ | $\alpha = 0.01$ | $\alpha = 0.1$ | $\alpha = 0.5$ |
| CIFAR-100 | ResNet-18 | *TCT* | **68.42%** | **69.07%** | **69.66%** | **69.68%** |
| | | *TCT-last-layer* | 59.80% | 60.04% | 64.98% | 66.22% |