# OpenReview forum: "TCT: Convexifying Federated Learning using Bootstrapped Neural Tangent Kernels"
_NeurIPS.cc/2022/Conference — NeurIPS 2022 Accept_

### Official Review · Reviewer_UZ2q · 2022-07-04

**Rating:** 6
**Confidence:** 4
**Soundness:** 3 good
**Presentation:** 3 good
**Contribution:** 2 fair

**Summary:**

This paper makes two main contributions: 1) empirically observing that the early layers of neural networks trained by FedAvg learn useful features even in heterogeneous data settings, while the performance degradation due to data heterogeneity is due to ineffective later layers, and 2) based on this observation, proposing a novel two-stage cross-silo federated learning method, BooNTK, that first uses FedAvg to learn early-layer features, applies a particular transformation based on these features to all data points, then learns a linear classifier on top of the transformed data using SCAFFOLD.

To verify that FedAvg learns useful early-layer features, FedAvg is run on a heterogeneous image dataset, then the last $\ell$-many layers are retrained in a centralized manner while the earlier layers are held fixed. It is observed that using the FedAvg-pretrained early layers leads to huge performance improvement over using random weights for the early layers when the last layer is retrained centrally. Also, there is some improvement due to using the FedAvg-pretrained vs random weights when the last $\ell$-many layers are retrained centrally for all $\ell$, suggesting that the early layers learned by FedAvg are extracting useful information. This observation inspires the first stage of the proposed method: FedAvg pretraining to learn the early layer weights. The paper also observes that SCAFFOLD is robust to data heterogeneity when the client losses are convex (but is not robust when the losses are nonconvex). This inspires the second stage of the proposed method: applying SCAFFOLD to convex losses to learn the last linear layer of the network. In particular, the second stage approach consists of computing the neural tangent kernel (NTK) representation of each data point using the pretrained weights in the NTK computation. Then, SCAFFOLD is employed to solve a multi-output linear regression with the transformed data points as inputs and the one-hot encoded labels as the target vectors. Empirical results are provided showing large improvement in training and testing accuracy of the proposed method over FedAvg, FedProx and SCAFFOLD on CIFAR100, FEMNIST, and MNIST with very heterogeneous partitions.


**Questions:**

Why is the proposed approach limited to the cross-silo setting, with a small number of clients? It seems to me that it should also work for settings with many clients. Or, if there are many clients and fewer samples per client, does locally optimizing the high-dimensional linear regression diverge?


**Limitations:**

The authors should discuss the computational complexity of the proposed method.

**Strengths And Weaknesses:**

Strengths

- The paper makes interesting empirical observations that are relevant to NeurIPS and may inspire future work. In particular the observation that FedAvg learns useful features even in data heterogeneous settings is novel and helps to explain the empirical success of FedAvg plus fine-tuning.

- These observations motivate a new federated learning method with promising empirical performance.

- The experimental evaluation is mostly thorough, with multiple datasets tested and helpful ablations.

- The writing is clear.

Weaknesses

1. Not enough intuition or empirical evidence is provided for why the second stage of the algorithm should consist of linear regression on the NTK-transformed data points rather than simply fixing the first L-1 layers and running lear regression to learn the paramters of the last layer with MSE loss, or learn it with multi-class logistic regression and cross entropy loss as is conventional. Both of the latter approaches maintain convexity of the loss functions, are much simpler to implement, and involve an optimization over far fewer parameters (presumably the output of the last layer has dimension far less than p=100,000 as in the NTK approach). These approaches should be compared against as baselines and intuition should be provided as to why they are not used.

2. On a similar note, results on the computational cost of the proposed method should be included. It seems to be much larger than the baselines due to mapping every data point to a high dimension via parameter derivative computation.

3. Personalized FL has been shown to be an effective alternative to learning a single global model in data heterogeneous settings. As such, some personalized FL methods should also be compared against. Also, the experimental results would be strengthened by comparison against more recent FL methods that learn a single global model, e.g. [23,63]. Comparing with only 3 baselines is low for an empirical paper.

Minor notes
- Missing related works: Huang et al., 2021 and Yue  et al., 2021 employ the NTK for FL.

- Intuition on why it makes sense to run linear regression for classification problems rather than logistic regression would be helpful.

- Since BootNTK gets to pretrain using FedAvg for T_1 rounds, fair comparison with the other methods should allow them to train for an extra T_1 rounds.

-Cross-device FL may still have data heterogeneity

- SCAFFOLD has higher communication and computational cost than fedavg by constant factor due to computing and communicating the gradient correction terms (footnote b).

- “By taking an eNTK approximation, BooNTK optimizes a convex approximation while using information from all layers of the model.” - besides the last layer

Huang et al., FL-NTK: A Neural Tangent Kernel-based Framework for Federated Learning Convergence Analysis, https://arxiv.org/pdf/2105.05001.pdf, 2021.
Yue et al., Neural Tangent Kernel Empowered Federated Learning, https://arxiv.org/pdf/2110.03681.pdf, 2021.

---

> ### Author Response · Authors · 2022-08-02
> **Response and thank you for your feedback (part 2)**
>
> >**Q4**: *Also, the experimental results would be strengthened by comparison against more recent FL methods that learn a single global model, e.g. [23,63]. Comparing with only 3 baselines is low for an empirical paper.*
>
> **A4**: In addition to the three baselines in Table 1, in our initial submission, we also compared our method to FedAdam [54] and FedDyn [1] in the appendix (Table 3 in Appendix B.1).
>
> Thank you for suggesting new baselines. FedNova [63] is mainly useful when the number of updates on the clients are very different, but has no effect when each client has a similar number of datapoints. In our most challenging setting of C=#1 and C=#2, all clients have a similar number of datapoints. As suggested, we have conducted new experiments on comparing with FedNova in Table 10, Appendix B.5 of our revised submission. As shown in Table 10, BooNTK significantly outperforms FedNova on CIFAR10/100 in highly non-iid settings. Regarding server momentum [23], FedAdam has been shown to outperform it in recent work [53]. Hence we chose to compare against FedAdam. We will include the results of FedNova and server momentum in our final version.
>
>
> >**Q5**: *Missing related works: Huang et al., 2021 and Yue et al., 2021 employ the NTK for FL.*
>
> **A5**: Thank you for pointing out these references. We have added [Huang et al., 2021] and [Yue et al., 2021] to Section 4.1 of our revised submission.
>
> >**Q6**: *Intuition on why it makes sense to run linear regression for classification problems rather than logistic regression would be helpful.*
>
> **A6**: Thank you for pointing this out. Linear regression is easier than logistic regression in federated learning because the Hessian remains constant [Woodworth et al. 2020]. Furthermore, quadratic loss may sometimes work better even in the centralized setting [Achille et al. 2021].
>
> Also, we have conducted new experiments on comparing the performance of quadratic loss and cross-entropy loss for BooNTK in Table 8, Appendix B.5 (BooNTK-CE) of our revised submission. As shown in Table 8, we find quadratic loss indeed achieves better performance than cross-entropy loss for BooNTK. We will include these results in our final version.
>
> [Woodworth et al. 2020] Woodworth, Blake E., Kumar Kshitij Patel, and Nati Srebro. "Minibatch vs local sgd for heterogeneous distributed learning." https://arxiv.org/abs/2006.04735 NeurIPS 202.
>
> [Achille et al. 2021] Achille, Alessandro, et al. "Lqf: Linear quadratic fine-tuning." https://arxiv.org/abs/2012.11140 CVPR 2021.
>
> >**Q7**: *Since BootNTK gets to pretrain using FedAvg for T_1 rounds, fair comparison with the other methods should allow them to train for an extra T_1 rounds.*
>
> **A7**: For all comparison experiments in our submission, we set the same number of communications rounds for our proposed method and existing methods. Specifically, we set T_1=T_2=100 for BooNTK on all datasets, and we run T_1 + T_2=200 rounds for existing methods. Further, we also compared the train/test curves of BooNTK and existing methods in Figure 6, Appendix B.3 in our initial submission. Thank you for pointing this out, and we will highlight and clarify this in our final version.
>
> >**Q8**: *SCAFFOLD has higher communication and computational cost than fedavg by constant factor due to computing and communicating the gradient correction terms (footnote b).*
>
> **A8**: As mentioned in Appendix A.1 of our initial submission, we describe a more communication efficient implementation of SCAFFOLD which is equivalent to Option II of SCAFFOLD [35]. Our implementation only requires a single model to be communicated between the client and server each round, making its communication complexity exactly equivalent to that of FedAvg.
>
> >**Q9**: *Or, if there are many clients and fewer samples per client, does locally optimizing the high-dimensional linear regression diverge?*
>
> **A9**: The linear regression should not be affected by small amounts of data. In fact, even in our setting, the linear model is overparameterized, with more parameters than data points per client.
>
> We also conducted new experiments on the large number of clients setting, where we considered the number of clients K=50 on CIFAR100 with alpha=0.001. The results are summarized in Table 11, Appendix B.5 of our revised submission. We find our proposed method (BooNTK: 45.32%) significantly outperforms existing methods (best: 16.70% test accuracy). Thank you for your suggestion on the large number of clients experiments, and we will include these results in our final version.

---

> > ### Comment · Reviewer_UZ2q · 2022-08-08
> > **Response**
> >
> > I thank the authors for their helpful and thorough response. Their clarifications and new experimental results (especially Table 9) have convinced me to raise my score. I would still like to see intuition on why eNTK representations perform better than the standard last-layer representations in future revisions.

---

> ### Author Response · Authors · 2022-08-02
> **Response and thank you for your feedback (part 1)**
>
> We thank the reviewers for their careful reading of our paper and help with improving our manuscript. We sincerely appreciate that you find our work proposes '*a novel two-stage scheme*' (**Reviewer rWVN**, **Reviewer UZ2q**) and '*computationally more efficient*' method (**Reviewer 7oaK**), provides '*promising empirical performance*' (**Reviewer UZ2q**) and '*significant improvement by the BooNTK pipeline on all tasks involving a varying amount of label skew*', '*intuitive ablations and method exposition*' (**Reviewer 7oaK**), and '*helpful ablations*' (**Reviewer UZ2q**), and '*may inspire future work'* (**Reviewer UZ2q**).
>
> In what follows, we try to address your concerns/questions and provide a detailed item-by-item response to your comments.
>
> ======================================================================================
>
> >**Q1**: *Not enough intuition or empirical evidence is provided for why the second stage of the algorithm should consist of linear regression on the NTK-transformed data points rather than simply fixing the first L-1 layers and running lear regression to learn the paramters of the last layer with MSE loss, or learn it with multi-class logistic regression and cross entropy loss as is conventional. These approaches should be compared against as baselines and intuition should be provided as to why they are not used.*
>
> **A1**: Thank you for your suggestion. We have conducted new experiments on using the last layer representations and cross-entropy loss in the Stage 2 of BooNTK, and the results are summarized in Table 9, Appendix B.5 of our revised submission. From Table 9, we find that applying eNTK representations with high dimension outperforms using the representations before the last layer only, especially in the settings with high degrees of data heterogeneity. These results provide further evidence on applying eNTK features instead of the representations before the last layer features. We will include these results in our final version.
>
> >**Q2**: On a similar note, results on the computational cost of the proposed method should be included. It seems to be much larger than the baselines due to mapping every data point to a high dimension via parameter derivative computation.
>
> **A2**: Thank you for pointing this out.
> In terms of communication cost, due to the subsampling step in BooNTK, BooNTK actually reduces 100x communication costs compared to existing methods on the CIFAR10 and CIFAR100 datasets (#parameter of the whole model: 11,169,345, #parameters of the subsample eNTK: 100,000).
> In terms of training, during Stage 2 of BooNTK, every client only needs to compute the eNTK representations once and use the computed representations to learn the linear model. The computing eNTK representations step is negligible compared to the overall training.
> In terms of inference, as pointed out in Q2 from Reviewer rWVN, the inference of our approach can be 3-5 times slower than standard with our method, and this can be a problem. However, 1) modern autograd libraries (e.g. JAX), and 2) computationally efficient approximations (see https://arxiv.org/abs/2004.0552) may significantly close this gap.
> We will include more results and discussions on the computation cost of BooNTK and speeding up inference of BooNTK for future work in our final version.
>
> >**Q3**: *Personalized FL has been shown to be an effective alternative to learning a single global model in data heterogeneous settings. As such, some personalized FL methods should also be compared against.*
>
> **A3**: Personalization requires that each client has a separate (possibly different) test dataset. Here, we investigate the setting where all clients want to do well on the “entire” test dataset, not just on their local dataset. E.g. suppose each client has a single class data (client k has class k). A personalized model which trivially always answers k has a perfect personalized accuracy. Here, we are interested in solving the more challenging problem of each client being able to classify all classes. See lines 25-30 for motivation.

---

### Official Review · Reviewer_7oaK · 2022-07-05

**Rating:** 6
**Confidence:** 4
**Soundness:** 2 fair
**Presentation:** 3 good
**Contribution:** 3 good

**Summary:**

This work proposes BooNTK; a two stage approach for FL that allows for better performance in cross-silo settings. The main idea is to 1) perform standard FedAvg training on a non-convex model, such as a neural network, then 2) approximate it with a first order Taylor approximation around the parameters found and finally 3) optimize the parameters of this linear model with federated training using optimisers with gradient correction, such as SCAFFOLD. The authors motivate this through the optimization difficulties of non-convex models in the non-iid setting; empirically, the loss of performance when moving to non-idd data is larger for non-convex model compared to convex.

The authors describe several further approximations that they do to make the method more efficient (i.e., removing the bias term of the Taylor approximation and only considering a subset of the coordinates of the gradient vector) and then demonstrate BooNTK’s performance on three label skew non-iid settings on FMNIST, CIFAR10 and CIFAR100.


**Questions:**

Most of my questions revolve around the weaknesses described above.

1. The authors do not take into account that the linear model is less flexible by design, therefore it is harder to fit the non-iid peculiarities (and could explain its improved performance). I believe a control experiment would highlight whether this can be an issue in practice. For example, one could do a heavily regularised non-convex model to see if the gap is shrinking in Figure 1. How would, e.g., the performance of a pipeline similar to BooNTK fare if instead of the linear model fitting step (i.e., stage 2), one just switched from FedAvg to FedProx with a strong regularisation strength for stage 2?

2. I believe that the effect of various sources of non-iidness on BooNTK should be investigated, as the experiments consider only label skew (where generally adjusting the classifier only is sufficient). What happens when there is, e.g., covariate shift? Intuitively, this should affect the earlier layers more.

3. I believe some more investigation on the eNTK part of BooNTK is required. For example, how does finetuning just the classification layer (while keeping the rest of the network frozen) with SCAFFOLD work, relative to the eNTK approach (which requires further approximations)? Furthermore, how does BooNTK perform without the approximations to the eNTK (i.e., do the approximations have beneficial regularisation effects or are they detrimental for performance)?

4. The authors consider classification tasks, however for the second stage of BooNTK, they consider an MSE loss on an one-hot representation of the targets. What is the motivation of the MSE loss? Intuitively, you should be able to apply a softmax on the linear model of the first eq. at sec. 4.1 to get a logistic regression model, which is more appropriate for classification.


**Limitations:**

The authors could have spent a bit more time discussing potentially negative aspects of their method (e.g., the linearity of the model in the second stage).

**Strengths And Weaknesses:**

Strengths
- Good results on the cross-silo setting
    - There is significant improvement by the BooNTK pipeline on all tasks involving a varying amount of label skew.
- Intuitive ablations and method exposition
    - The method is explained / motivated well and I liked the layer importance investigation. The ablation studies are also useful and highlight the sensitivity of the method on the choice of (some) hyper parameters.
- Simple method
    - The method is quite simple and straightforward. As a bonus, the second step is also computationally more efficient than training the original neural network.

Weaknesses:
- No discussion about how different non-iid ness settings affect the method
    - Given the claims of the work about the negative effects of data heterogeneity in the non-convex setting, I would have expected that the authors experimented with more diverse non-iid settings (i.e., not just label skew). As the current label-skew experiments concern mostly different marginals over the labels, $p(y)$, at each client, a better adjusted classification layer is important. I believe this is one of the reasons that BooNTK is effective in these scenarios. However, not all non-iidness is just label skew; for example, you could consider an image recognition system running in different mobile phones. As each phone comes with its own camera sensor, covariate shift (i.e., different $p(x)$ across clients) can also manifest as a source of non-iidness. In this case, I would intuitively expect that the distribution of the features also differs among clients, therefore, adjusting just the classifier might not be enough.
- No discussion about how the linear model is less flexible
    - The linear model is less flexible than the non-convex neural network. As a result, its improved performance (on a given feature set) could be just because it has less degrees of freedom to adapt to the non-i.i.d. peculiarities. I believe a discussion around this is missing.
- There are several approximations involved in the computation of the eNTK features, the impact of which is unclear
    - In section 4.1 the authors describe a series of approximations to eNTK in order to reduce the computational burden, however there is no (empirical) evaluation on how each one affects the final performance.

---

> ### Author Response · Authors · 2022-08-02
> **Response and thank you for your feedback (part 2)**
>
> >**Q4**: *How would, e.g., the performance of a pipeline similar to BooNTK fare if instead of the linear model fitting step (i.e., stage 2), one just switched from FedAvg to FedProx with a strong regularisation strength for stage 2?*
>
> **A4**: Thank you for your suggestion on the experiments. We have conducted new experiments on switching from FedAvg to FedProx on CIFAR10 (#C=2). By sweep over the the regularization parameter from {10.0, 1.0, 0.1, 0.01, 0.001} for the FedProx (after switch from FedAvg), the best performance is 58.74%, which is only slightly better than FedAvg (56.86%) and FedProx (56.87%) and much worse than BooNTK (83.02%). These results suggest that the gain is not due to the regularization.
>
>
> >**Q5**: *The authors do not take into account that the linear model is less flexible by design, therefore it is harder to fit the non-iid peculiarities.*
>
> **A5**: We disagree with the argument ‘it is harder to fit the non-iid peculiarities’. As shown in Figure 2(c), Figure 5, and Figure 6, the train accuracy of BooNTK is almost 100% training accuracy across different settings.
>
> >**Q6**: How does finetuning just the classification layer (while keeping the rest of the network frozen) with SCAFFOLD work, relative to the eNTK approach (which requires further approximations)?
>
> **A6**: Thank you for your suggestions on experiments. We have conducted new experiments on using the last layer representations and cross-entropy loss in the Stage 2 of BooNTK, and the results are summarized in Table 9, Appendix B.5 of our revised submission. From Table 9, we find that applying eNTK representations with high dimension outperforms using the representations before the last layer only, especially in the settings with high degrees of data heterogeneity. These results provide further evidence on applying eNTK features instead of the representations before the last layer features. We will include these results in our final version.
>
>
> >**Q7**: The authors consider classification tasks, however for the second stage of BooNTK, they consider an MSE loss on an one-hot representation of the targets. What is the motivation of the MSE loss? Intuitively, you should be able to apply a softmax on the linear model of the first eq. at sec. 4.1 to get a logistic regression model, which is more appropriate for classification.
>
> **A7**: Thank you for pointing this out. Linear regression is easier than logistic regression in federated learning because the Hessian remains constant [Woodworth et al. 2020]. Furthermore, quadratic loss may sometimes work better even in the centralized setting [Achille et al. 2021].
>
> Also, we have conducted new experiments on comparing the performance of quadratic loss and cross-entropy loss for BooNTK in Table 8, Appendix B.5 (BooNTK-CE) of our revised submission. As shown in Table 8, we find quadratic loss indeed achieves better performance than cross-entropy loss for BooNTK. We will include these results in our final version.
>
> [Woodworth et al. 2020] Woodworth, Blake E., Kumar Kshitij Patel, and Nati Srebro. "Minibatch vs local sgd for heterogeneous distributed learning." https://arxiv.org/abs/2006.04735 NeurIPS 202.
>
> [Achille et al. 2021] Achille, Alessandro, et al. "Lqf: Linear quadratic fine-tuning." https://arxiv.org/abs/2012.11140 CVPR 2021.

---

> > ### Comment · Reviewer_7oaK · 2022-08-08
> > **Response to rebuttal**
> >
> > I would like to thank the authors for their extensive response that cleared a lot of my concerns. I would also encourage the authors to provide the discussion around the sources of non-iidness in the main text as well. Based on the above, I will raise my score.

---

> ### Author Response · Authors · 2022-08-02
> **Response and thank you for your feedback (part 1)**
>
> We thank the reviewers for their careful reading of our paper and help with improving our manuscript. We sincerely appreciate that you find our work proposes '*a novel two-stage scheme*' (**Reviewer rWVN**, **Reviewer UZ2q**) and '*computationally more efficient*' method (**Reviewer 7oaK**), provides '*promising empirical performance*' (**Reviewer UZ2q**) and '*significant improvement by the BooNTK pipeline on all tasks involving a varying amount of label skew*', '*intuitive ablations and method exposition*' (**Reviewer 7oaK**), and '*helpful ablations*' (**Reviewer UZ2q**), and '*may inspire future work'* (**Reviewer UZ2q**).
>
> In what follows, we try to address your concerns/questions and provide a detailed item-by-item response to your comments.
>
> ======================================================================================
>
> >**Q1**: No discussion about how different non-iid ness settings affect the method. However, not all non-iidness is just label skew; for example, you could consider an image recognition system running in different mobile phones. As each phone comes with its own camera sensor, covariate shift (i.e., different $p(x)$ across clients) can also manifest as a source of non-iidness.
>
> **A1**: Past research (e.g. [Li et al. 2021]) has shown that label skew is the most challenging form of heterogeneity, and hence we focused on this in our work. Further, label skew automatically results in all kinds of covariate shifts since the classes naturally have very different features. For example, the different classes in CIFAR10/100 have different colors, textures, and camera technologies used (refer to Figure 7, Appendix B.5 for visually comparing pictures of ‘cat’ with ‘ship’ in CIFAR10). Having said this, we agree that there are other interesting notions of covariate shift whose effect we do not explore. This is a strong limitation of current evaluation methods in federated learning in general - we need new real world datasets with more interesting and realistic heterogeneities in order to carry out such investigations.
>
> [Li et al. 2021] Qinbin Li, Yiqun Diao, Quan Chen, and Bingsheng He. Federated learning on non-iid data silos: An experimental study. arXiv preprint arXiv:2102.02079, 2021
>
>
> >**Q2**: *No discussion about how the linear model is less flexible. As a result, its improved performance (on a given feature set) could be just because it has less degrees of freedom to adapt to the non-i.i.d. peculiarities.*
>
> **A2**: Thank you for your suggestion on discussing the limitation of the linear model. We will include additional discussions on limitations of the linear model part in our final version.
>
> We politely disagree with the reasoning of your argument ‘its improved performance could be just because it has less degrees of freedom to adapt to the non-i.i.d. peculiarities’. The linear model has just as many degrees of freedom as the neural network since it has the same number of parameters. In fact, as shown in Figure 2(c), Figure 5, and Figure 6, the train accuracy of BooNTK is almost 100% training across different settings, which means that our gain cannot be because of this.
>
> >**Q3**: *There are several approximations involved in the computation of the eNTK features, the impact of which is unclear. In section 4.1 the authors describe a series of approximations to eNTK in order to reduce the computational burden, however there is no (empirical) evaluation on how each one affects the final performance. Furthermore, how does BooNTK perform without the approximations to the eNTK (i.e., do the approximations have beneficial regularisation effects or are they detrimental for performance)?*
>
> **A3**: Thank you for your suggestion. We described two approximations in Section 4.1: (1). randomly reinitialize the last layer and only consider with respect to a single output logit; (2). subsample random coordinates from the eNTK feature. As explained in Line 150 - Line 157 in Section 4.1, the first approximation does not sacrifice the representation power.
>
> As suggested, we have conducted new experiments on applying full eNTK representation in BooNTK - Stage 2 (without random sampling) to investigate the role of the second approximation. We provide the results in Table 8, Appendix B.5 (BooNTK-full-eNTK). As shown in Table 8, applying full eNTK representations slightly improves (improvements are smaller than 2% across all settings) the performance of BooNTK on CIFAR10/100. On the other hand, using subsampled eNTK *reduces the communication cost more than 100x* compared to the full eNTK and existing federated learning algorithms (#parameter of the whole model: 11,169,345, #parameters of the subsample eNTK: 100,000). We will include the full eNTK results as well as additional discussions on the communication costs in our final version.

---

### Official Review · Reviewer_rWVN · 2022-07-09

**Rating:** 6
**Confidence:** 4
**Soundness:** 3 good
**Presentation:** 2 fair
**Contribution:** 2 fair

**Summary:**

This paper focuses on difficulty introduced by non-convexity and data heterogeneity in federated learning.
Authors first show that, with data heterogeneity, linear models and convex optimization problem can be trained efficiently with gradient correction techniques such as SCAFFOLD, while non convex problem cannot.
Then, in order to sidestep the non-convexity for neural network, authors substitute original model with a linear approximation and original loss with a quadratic loss, such that non-convex optimization problem turns into a convex regression problem based on NTK.
Since feature learning is necessary for NTK align with data to provide good result, authors introduce two-stage training, where feature learning only happens in first stage, and convex optimization with gradient correction happens in second stage.
Authors conduct experiments to show that such two-stage optimization can fit the data faster and produce better test accuracy.

**Questions:**

If I understand correctly, the final training result for BookNTK is a neural network plus a linear model. Any inference requires a backward propagation to derive the feature for empirical NTK to apply a linear model. This is different from common workflow. Is there a way to compile the final network and the linear model into a single model that support inference with a single forward pass?

Section 3 mentioned "the train and test accuracies in Figure 1(b) match quite closely, suggesting that the failure lies in optimization". However Figure 2 shows a different picture, where train acc and test acc have a large gap. What's the reason behind this discrepancy?

Why quadratic loss instead of cross-entropy is used in (1)?

How much worse is BookNTK than centralized training? Authors only mentioned one centralized baseline in section 3, where they show that "21% (out of the 35% gap in accuracy) may be attributed to a failure to optimize the linear output layer". However, for BookNTK, none of results in section 5 and appendix contains centralized training. I think this question is as important as "How much better is BookNTK than FedAvg". I strongly suggest authors to add centralized training result for a comparison.

**Limitations:**

Yes.

**Strengths And Weaknesses:**

Strength:

This paper introduced a novel two-stage scheme to combine the feature learning capacity of neural network and for efficient optimization linear model.

Weakness:

The convexified problem is introduced mainly due to an optimization consideration: SCAFFOLD perform well on convex method. However, the convex formulation clearly sacrifice the model capacity. The feature learning can only happen in first stage (or non-convex part), thus BookNTK learns less feature than centralized training, which doesn't involve two-stage training.

---

> ### Author Response · Authors · 2022-08-02
> **Response and thank you for your feedback (part 2)**
>
> >**Q4**: Why quadratic loss instead of cross-entropy is used in (1)?
>
> **A4**: Thank you for pointing this out. Linear regression is easier than logistic regression in federated learning because the Hessian remains constant [Woodworth et al. 2020]. Furthermore, quadratic loss may sometimes work better even in the centralized setting [Achille et al. 2021].
>
> Also, we have conducted new experiments on comparing the performance of quadratic loss and cross-entropy loss for BooNTK in Table 8, Appendix B.5 (BooNTK-CE) of our revised submission. As shown in Table 8, we find quadratic loss indeed achieves better performance than cross-entropy loss for BooNTK. We will include these results in our final version.
>
> [Woodworth et al. 2020] Woodworth, Blake E., Kumar Kshitij Patel, and Nati Srebro. "Minibatch vs local sgd for heterogeneous distributed learning." https://arxiv.org/abs/2006.04735 NeurIPS 202.
>
> [Achille et al. 2021] Achille, Alessandro, et al. "Lqf: Linear quadratic fine-tuning." https://arxiv.org/abs/2012.11140 CVPR 2021.
>
> >**Q5**: *How much worse is BookNTK than centralized training? I strongly suggest authors to add centralized training result for a comparison.*
>
> **A5**: Thank you for your great suggestion. We have added the results of centralized training to our main comparison table (Table 2 in Section 5) in our revised submission. Overall, BooNTK performs worse than the centralized training, especially in the highly non-iid setting. However, the gaps between BooNTK and centralized training are much smaller than existing methods. We will include the centralized training results in our final version.

---

> ### Author Response · Authors · 2022-08-02
> **Response and thank you for your feedback (part 1)**
>
> We thank the reviewers for their careful reading of our paper and help with improving our manuscript. We sincerely appreciate that you find our work proposes '*a novel two-stage scheme*' (**Reviewer rWVN**, **Reviewer UZ2q**) and '*computationally more efficient*' method (**Reviewer 7oaK**), provides '*promising empirical performance*' (**Reviewer UZ2q**) and '*significant improvement by the BooNTK pipeline on all tasks involving a varying amount of label skew*', '*intuitive ablations and method exposition*' (**Reviewer 7oaK**), and '*helpful ablations*' (**Reviewer UZ2q**), and '*may inspire future work'* (**Reviewer UZ2q**).
>
> In what follows, we try to address your concerns/questions and provide a detailed item-by-item response to your comments.
>
> ======================================================================================
>
> >**Q1**: *The convex formulation clearly sacrifice the model capacity. The feature learning can only happen in first stage (or non-convex part), thus BookNTK learns less feature than centralized training, which doesn't involve two-stage training.*
>
> **A1**: This is a good point, as you say, the second phase of BooNTK does not perform feature learning and only the first phase of BooNTK performs feature learning. However, as shown in Figure 6, Appendix B.3, we study the performance of BooNTK with a different number of rounds ($T_1$) for phase1. We find that using larger $T_1$ does not significantly improve the test accuracy of BooNTK when $T_1$ is larger than 60, which suggests that the feature learning saturates after ~60 communication rounds in phase 1 of BooNTK. Moreover, BooNTK leverages the effective features learned in phase 1 and significantly improves the model performance compared to existing approaches.
>
> Meanwhile, our convex formulation does not sacrifice the model capacity. For example, BooNTK achieves ~100% training accuracy in phase 2, as shown in Figure 5 and Figure 6. In contrast, BooNTK aims to maximize the usage of the features learned in phase 1 by using more features (i.e., overparameterized linear model).
>
> Lastly, as shown in our experimental results, BooNTK achieves fast convergence in Stage 2 (less than 20 communication rounds). This suggests that our proposed convexify approach could serve as an effective approach, that is complementary to existing methods, for tackling the heterogeneity issue in federated learning.
>
> >**Q2**: *If I understand correctly, the final training result for BookNTK is a neural network plus a linear model. Is there a way to compile the final network and the linear model into a single model that support inference with a single forward pass?*
>
> **A2**:  Yes, this is correct. Indeed, the inference of our approach can be 3-5 times slower than standard with our method, and this can be a problem. However, 1) modern autograd libraries (e.g. JAX), and 2) computationally efficient approximations (see https://arxiv.org/abs/2004.0552) may significantly close this gap. We will include more discussions on speeding up our proposed approach in our final version.
>
> >**Q3**: *Section 3 mentioned "the train and test accuracies in Figure 1(b) match quite closely, suggesting that the failure lies in optimization". However Figure 2 shows a different picture, where train acc and test acc have a large gap. What's the reason behind this discrepancy?*
>
> **A3**: Thank you for pointing this out. FedAvg and SCAFFOLD show a larger gap only in the easy nearly-iid setting, where training is not a big issue. The setting in Figure 1(b) – where each client only has samples from 2 classes – has a much higher degree of non-iid-ness than the ones in Figure 1(b). We will incorporate your suggestion and improve the clarity of our presentation in our final version.

---

> > ### Comment · Reviewer_rWVN · 2022-08-08
> > **Thanks for the Response**
> >
> > I thank the authors for their response, which address most of my concerns.
> >
> > The only remaining question is Q3.
> > Authors state that "FedAvg and SCAFFOLD show a larger gap only in the easy nearly-iid setting". However, my original question is regarding to gap between train and test curve.
> >
> > In Figure 1 (b), the curve `FedAvg-Train ($\alpha=0.1$)` is very close to `FedAvg-Test ($\alpha=0.1$)`. However, In Figure 2 (a), the training and testing curve with $\alpha=0.1$ for FedAvg have a large gap.
> > The only difference between Figure 1 (b) and  Figure 2 (a) I noticed so far is they use different dataset. However, how does it affect the gap between training and testing curve? Could any argument similar to lines 118-120 be given for Figure 2 (a)?

---

> > > ### Author Response · Authors · 2022-08-08
> > > **Thank you for the reply and response to Q3**
> > >
> > > We thank you for your feedback!
> > >
> > > Regarding question **Q3**: Apologies for misunderstanding the previous question. Fig 1 uses CIFAR10 whereas Fig 2 uses CIFAR100. The latter is more challenging and is known to have a bigger generalization gap even in the centralized setting. In all cases, the generalization gap in our federated setting closely matches those seen in the centralized setting.
> > >
> > > In the centralized setting, ResNet on CIFAR100 has a ~25% generalization gap between train and test accuracy, whereas on CIFAR10 it is ~8%. Similar generalization gaps are observed in our federated experiments as well, except that both the train and the test accuracies are shifted down due to poor optimization.
> > >
> > > *Since the gap is not significantly larger in the federated setting, we can conclude the added difficulty is not due to generalization but due to optimization.* We will further revise lines 118-120 in our submission to clarify the dataset setup.
> > >
> > > Thank you again for your feedback. We hope this clarifies the question. Please let us know if you would like any more details about this.

---

> > > > ### Comment · Reviewer_rWVN · 2022-08-08
> > > > **Thanks for the response**
> > > >
> > > > Authors response and new experiments have resolved my concerns. Thus I will raise my score.

---

### Author Response · Authors · 2022-08-02
**Summary of changes and thank all the reviewers!**

We thank the reviewers for their careful reading of our paper and help with improving our manuscript. We sincerely appreciate that you find our work proposes '*a novel two-stage scheme*' (**Reviewer rWVN**, **Reviewer UZ2q**) and '*computationally more efficient*' method (**Reviewer 7oaK**), provides '*promising empirical performance*' (**Reviewer UZ2q**) and '*significant improvement by the BooNTK pipeline on all tasks involving a varying amount of label skew*', '*intuitive ablations and method exposition*' (**Reviewer 7oaK**), and '*helpful ablations*' (**Reviewer UZ2q**), and '*may inspire future work'* (**Reviewer UZ2q**).

We have modified our submission based on the suggestions/questions of the reviewers and have uploaded a revised version. Revised places are marked in blue color. In particular, we have made the following main updates:

1. Added the centralized training results (in Table 1, Section 5).

2. Added the results of BooNTK using full eNTK representations, i.e., without performing dimension reduction subsampling (in Table 8, Appendix B.5).

3. Added the results of BooNTK with cross-entropy loss used in Stage 2 (in Table 8, Appendix B.5).

4. Added the results of BooNTK using the representations before the last layer and cross-entropy loss (in Table 9, Appendix B.5).

5. Compared with FedNova and add the comparison results to Table 10 (Appendix B.5).

6. Added the comparison of the performance of BooNTK and existing methods when the number of clients is large in Table 11 (Appendix B.5).

---

### Author Response · Authors · 2022-08-08
**Looking for feedback/update from reviewers**

We thank you again for your thoughtful review and comments. With the author-reviewer period ending soon, we just wanted to reach out and see if any of the reviewers had any comments back to our rebuttal. We are looking for feedback on whether the points made in the reviews have now been addressed. We are happy to answer any remaining questions regarding our rebuttal or the paper itself. Thank you!

---

### Meta-Review · Area_Chair_pJCW · 2022-08-25

**Recommendation:** Accept
**Confidence:** Certain

**Metareview:**

This paper introduced a novel two-stage scheme to combine the feature learning capacity of neural network and for efficient optimization linear model. It makes interesting empirical observations that are relevant to NeurIPS and may inspire future work. In particular, the observation that FedAvg learns useful features even in data heterogeneous settings is novel and helps to explain the empirical success of FedAvg plus fine-tuning. The experimental evaluation is mostly thorough, with multiple datasets tested and helpful ablations.


**Award:**

No

---

### Decision · Program_Chairs · 2022-09-14

Accept